# A tudor domain protein, SIMR-1, promotes siRNA production at piRNA-targeted mRNAs in *C. elegans*

Kevin I Manage[1], Alicia K Rogers[1], Dylan C Wallis[1], Celja J Uebel[1], Dorian C Anderson[1], Dieu An H Nguyen[1], Katerina Arca[1], Kristen C Brown[2,3], Ricardo J Cordeiro Rodrigues[4,5], Bruno FM de Albuquerque[4], René F Ketting[4], Taiowa A Montgomery[2], Carolyn Marie Phillips[1]*

[1]Department of Biological Sciences, University of Southern California, Los Angeles, United States; [2]Department of Biology, Colorado State University, Fort Collins, United States; [3]Cell and Molecular Biology Program, Colorado State University, Fort Collins, United States; [4]Biology of Non-coding RNA Group, Institute of Molecular Biology, Mainz, Germany; [5]International PhD Programme on Gene Regulation, Epigenetics, and Genome Stability, Mainz, Germany

**Abstract** piRNAs play a critical role in the regulation of transposons and other germline genes. In *Caenorhabditis elegans*, regulation of piRNA target genes is mediated by the *mutator* complex, which synthesizes high levels of siRNAs through the activity of an RNA-dependent RNA polymerase. However, the steps between mRNA recognition by the piRNA pathway and siRNA amplification by the *mutator* complex are unknown. Here, we identify the Tudor domain protein, SIMR-1, as acting downstream of piRNA production and upstream of *mutator* complex-dependent siRNA biogenesis. Interestingly, SIMR-1 also localizes to distinct subcellular foci adjacent to P granules and *Mutator* foci, two phase-separated condensates that are the sites of piRNA-dependent mRNA recognition and *mutator* complex-dependent siRNA amplification, respectively. Thus, our data suggests a role for multiple perinuclear condensates in organizing the piRNA pathway and promoting mRNA regulation by the *mutator* complex.

*For correspondence: cphil@usc.edu

Competing interests: The authors declare that no competing interests exist.

## Introduction

In many eukaryotes, small RNAs, ranging from ~18–30 nucleotides in length, regulate cellular mRNAs through sequence complementarity. Argonaute proteins are key mediators of RNA silencing; by binding to small RNAs, which interact with fully or partially complementary mRNAs, the Argonaute proteins can promote transcription repression, translation inhibition, and RNA decay of targeted mRNAs (*Hutvagner and Simard, 2008*; *Claycomb, 2014*). Through this regulation of both endogenous and foreign RNAs, small RNAs play key roles in maintaining proper gene expression and silencing deleterious RNAs (*Claycomb, 2014*; *Ketting, 2011*).

A subclass of small RNAs, known as piRNAs, is critical for germ cell function, including silencing of transposons and other germline mRNAs (*Ketting, 2011*; *Weick and Miska, 2014*). piRNAs are bound by a subgroup of Argonaute proteins called Piwi proteins, of which *C. elegans* has a single functional homolog, PRG-1 (*Batista et al., 2008*; *Das et al., 2008*; *Wang and Reinke, 2008*). In many organisms, including mammals, flies, and zebrafish, piRNAs are amplified through the ping-pong mechanism (*Brennecke et al., 2007*; *Aravin et al., 2007*; *Gunawardane et al., 2007*; *Houwing et al., 2007*). This mechanism, however, is not found in nematodes. Rather, *C. elegans* employs a different mechanism to reinforce silencing at piRNA target loci. In *C. elegans*, a small RNA amplification pathway dependent on the *mutator* complex, which includes an RNA-dependent

**eLife digest** In the biological world, a process known as RNA interference helps cells to switch genes on and off and to defend themselves against harmful genetic material. This mechanism works by deactivating RNA sequences, the molecular templates cells can use to create proteins.

Overall, RNA interference relies on the cell creating small RNA molecules that can target and inhibit the harmful RNA sequences that need to be silenced. More precisely, in round worms such as *Caenorhabditis elegans*, RNA interference happens in two steps. First, primary small RNAs identify the target sequences, which are then combatted by newly synthetised, secondary small RNAs. A number of proteins are also involved in both steps of the process.

RNA interference is particularly important to preserve fertility, guarding sex cells against 'rogue' segments of genetic information that could be passed on to the next generation. In future sex cells, the proteins involved in RNA interference cluster together, forming a structure called a germ granule. Yet, little is known about the roles and identity of these proteins.

To fill this knowledge gap, Manage et al. focused on the second stage of the RNA interference pathway in the germ granules of *C. elegans*, examining the molecules that physically interact with a key protein. This work revealed a new protein called SIMR-1.

Looking into the role of SIMR-1 showed that the protein is required to amplify secondary small RNAs, but not to identify target sequences. However, it only promotes the creation of secondary small RNAs if a specific subtype of primary small RNAs have recognized the target RNAs for silencing.

Further experiments also showed that within the germ granule, SIMR-1 is present in a separate substructure different from any compartment previously identified. This suggests that each substep of the RNA interference process takes place at a different location in the granule.

In both *C. elegans* and humans, disruptions in the RNA interference pathway can lead to conditions such as cancer or infertility. Dissecting the roles of the proteins involved in this process in roundworms may help to better grasp how this process unfolds in mammals, and how it could be corrected in the case of disease.

RNA polymerase, synthesizes secondary downstream siRNAs from piRNA-targeted mRNAs to trigger robust and heritable silencing (*Das et al., 2008*; *Lee et al., 2012*; *Bagijn et al., 2012*; *Shirayama et al., 2012*; *Ashe et al., 2012*). These siRNAs are approximately 22-nt long, often start with a 5'G, and are bound by the WAGO clade of Argonaute proteins, including WAGO-1, therefore, they are often referred to as WAGO-class 22G-siRNAs (*Pak and Fire, 2007*; *Sijen et al., 2007*; *Yigit et al., 2006*; *Gu et al., 2009*).

In addition to the Piwi proteins, a major player in the piRNA pathway is the Tudor domain protein family. Tudor domain proteins in many organisms, including both mouse and *Drosophila,* play critical roles in piRNA accumulation and mRNA target regulation through their interaction with PIWI proteins (*Reuter et al., 2009*; *Chen et al., 2011*; *Nishida et al., 2009*). The Tudor domain is a conserved structural motif originally identified in the *Drosophila* protein Tudor (*Boswell and Mahowald, 1985*; *Ponting, 1997*; *Callebaut and Mornon, 1997*). Tudor domains, which function as protein-protein interaction modules, recognize methylated arginines or lysines and thus can mediate protein interactions in a methylation-specific manner (*Friesen et al., 2001*; *Chen et al., 2011*). Most often, methylarginine-binding Tudor domain proteins are associated with RNA metabolism, while methyllysine-binding Tudor domain proteins are involved in chromatin biology (*Chen et al., 2011*). Interestingly, Tudor domain proteins affiliated with the piRNA pathway often interact with an additional conserved element flanking the Tudor domain core referred to as the extended Tudor domain, which is required for their ability to recognize peptides containing a methylated arginine modification (*Chen et al., 2011*; *Liu et al., 2010a*; *Liu et al., 2010b*). The extended Tudor domain preferentially recognizes symmetrically dimethylated arginine (sDMA) modifications over monomethylated arginines (MMA), asymmetrically dimethylated arginines (aDMA), or unmodified peptides; however, some extended Tudor domain proteins have lost the ability to bind the methylated arginine mark and recognize only unmodified peptides (*Liu et al., 2010b*; *Zhang et al., 2017*). These arginine methylation modifications are often found within the context of arginine-glycine (RG) and arginine-

alanine (RA) repeats and are catalyzed by the activity of Protein Arginine Methyl Transferases (PRMTs) (*Kirino et al., 2009*; *Vagin et al., 2009*; *Reuter et al., 2009*; *Webster et al., 2015*; *Liu et al., 2010a*; *Nishida et al., 2009*).

Many components of the piRNA pathway, including some Piwi and Tudor domain proteins, are localized to membrane-less, cytoplasmic compartments at the periphery of germline nuclei. In *Drosophila,* the piRNA pathway components localize to a compartment referred to as nuage, and in *C. elegans,* these components localize to the P granule. Seminal work in *C. elegans* has shown that P granules assemble by intracellular phase separation (*Brangwynne et al., 2009*). More recently, both *Mutator* foci, the sites of secondary siRNA biogenesis by the *mutator* complex, and Z granules, which are required for RNAi inheritance, have been shown to be phase-separated biomolecular condensates which lie adjacent to one another and the P granule at the nuclear periphery (*Uebel et al., 2018*; *Wan et al., 2018*). This assembly of condensates can be referred to as PZM granules or as nuage. These discoveries have led to an intriguing model where the small RNA pathway is temporally and spatially organized into membrane-less organelles, with distinct steps of the silencing pathway occurring in neighboring condensates, while still allowing for trafficking of RNAs and perhaps some proteins between condensates.

Here we identify a protein required to coordinate RNA silencing between the piRNA pathway in P granules and siRNA amplification in *Mutator* foci. Specifically, through proteomic analysis of MUT-16, we identified an uncharacterized Tudor domain protein, SIMR-1 (**si**RNA-defective and **m**o**r**tal germline). Unlike *mut-16* mutants, *simr-1* mutants are not defective in exogenous RNAi, but do have a transgenerational sterility phenotype at elevated temperature. Interestingly, while SIMR-1 is not required for production of piRNAs or the expression of PRG-1, *simr-1* mutants fail to produce high levels of siRNAs from many piRNA-target loci. These data suggest that SIMR-1 may act at a step in between PRG-1 targeting and siRNA biogenesis by the *mutator* complex. Finally, we demonstrate that SIMR-1 localizes to perinuclear foci, adjacent to, but distinct from *Mutator* foci, P granules and Z granules, which we name SIMR foci. Therefore, this work identifies SIMR-1 as a factor that acts downstream of PRG-1 to mediate the production of secondary siRNAs by the *mutator* complex, and suggests a role for multiple perinuclear condensates to promote mRNA regulation by the piRNA pathway and *mutator* complex.

## Results

### Identification of MUT-16-associated proteins by functional proteomics

Many components of the *mutator* complex have been identified through forward and reverse genetic screens (*Supplementary file 1*; *Ketting and Plasterk, 2000*; *Ketting et al., 1999*; *Tabara et al., 1999*; *Vastenhouw et al., 2003*). More recently, three Zc3h12a ribonuclease-like proteins that interact with the *mutator* complex were identified through co-immunoprecipitation followed by mass spectrometry (IP-mass spec) (*Tsai et al., 2015*). We sought to take a similar approach and extend the list of *mutator* complex proteins and proteins that interact with the *mutator* complex. Because MUT-16 is a scaffolding protein required for assembly of the *mutator* complex (*Phillips et al., 2012*), we chose to use an endogenously tagged MUT-16::GFP::3xFLAG for immunoprecipitation. Following separate immunoprecipitations with GFP and FLAG antibodies and mass spectrometry analyses, we limited our candidate list to proteins that were present in both MUT-16-GFP and MUT-16-FLAG immunoprecipitations and absent in both wild-type immunoprecipitations. In total, we identified 17 candidate MUT-16 interactors, twelve of which comprise all known members of the *mutator* complex (*Phillips et al., 2012*; *Uebel et al., 2018*; *Tsai et al., 2015*) and five previously uncharacterized proteins (*Figure 1A* and *Supplementary file 2*). We additionally chose to further examine three proteins (RSD-2, WAGO-1, and MATH-33) that were present in the MUT-16-GFP immunoprecipitation, absent in the control GFP immunoprecipitation, and enriched at least four-fold in the MUT-16-FLAG immunoprecipitations relative to the control FLAG immunoprecipitation (*Figure 1—figure supplement 1A* and *Supplementary file 2*). RSD-2 is a small RNA factor required for exogenous RNAi introduced at low doses and not previously known to interact with the *mutator* complex (*Sakaguchi et al., 2014*; *Han et al., 2008*; *Tijsterman et al., 2004*; *Zhang et al., 2012*); WAGO-1 is an Argonaute protein that localizes to P granules but was found to interact with MUT-16 in a yeast two-hybrid screen (*Supplementary file 1*; *Gu et al., 2009*; *Phillips et al., 2014*);

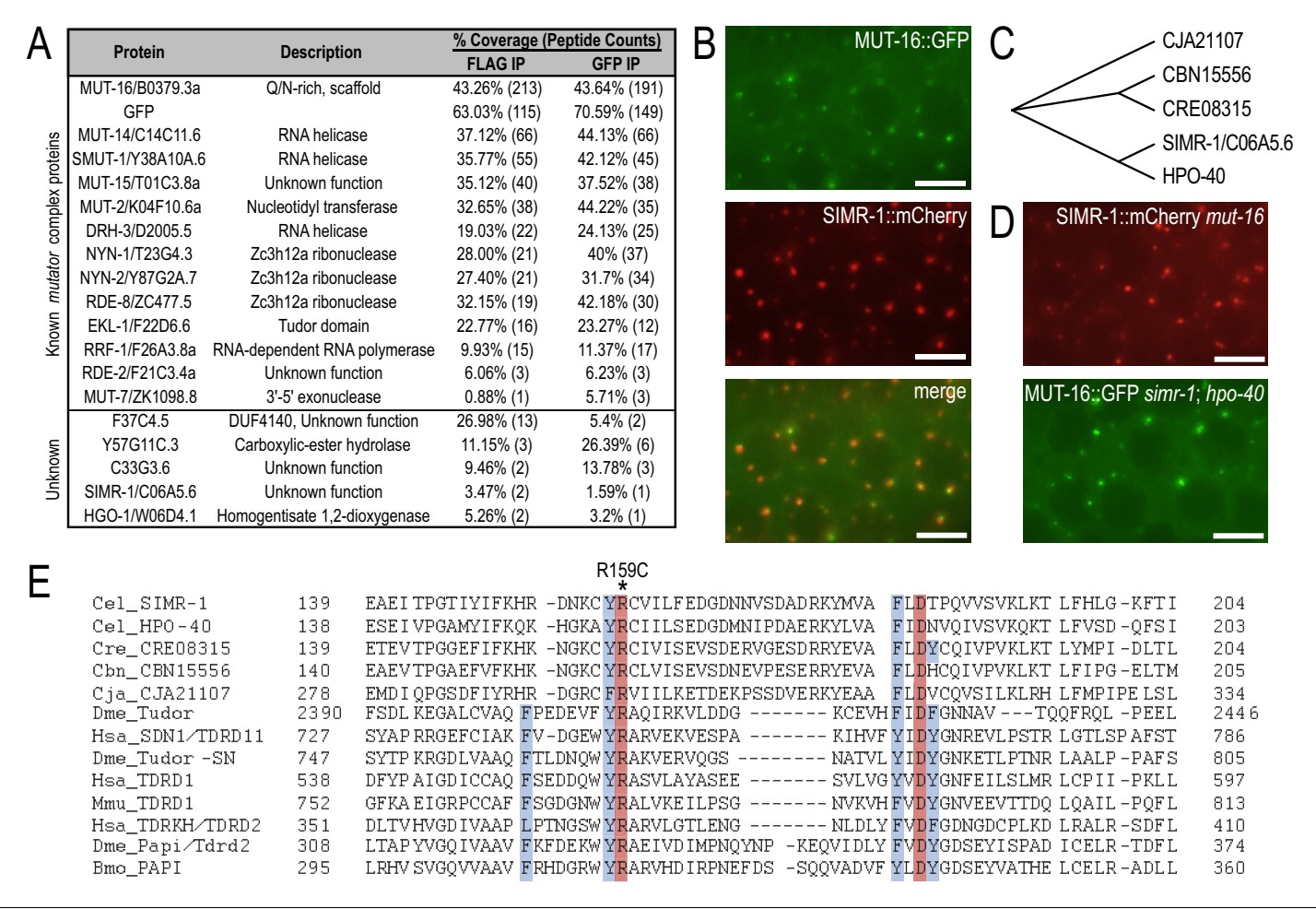

**Figure 1.** SIMR-1 is a perinuclear-localized Tudor domain protein. (A) Proteins identified by IP-mass spec of MUT-16::GFP::3xFLAG but not wild-type animals. The percent coverage and total number of peptides captured are indicated for each MUT-16-associated protein. See *Supplementary file 2* for complete list of immunoprecipitated proteins. (B) Live imaging of SIMR-1::mCherry demonstrate that it is adjacent to or colocalizes with MUT-16::GFP foci. Scale bars, 5 μm. (C) Cladogram representing the relationship between SIMR-1 and related proteins CJA21107 (*C. japonica*), CBN15556 (*C. brenneri*), CRE08315 (*C. remanei*), and HPO-40 (*C. elegans*). The protein alignment was generated using Clustal Omega and cladogram was made in Evolview V3. (D) Live imaging of SIMR-1::mCherry in a *mut-16* mutant and MUT-16::GFP in a *simr-1; hpo-40* double mutant indicate that *mut-16* is not required for SIMR-1 foci formation, nor are *simr-1* and *hpo-40* required for *Mutator* foci formation. Scale bars, 5 μm. (E) Alignment of Tudor domain region generated by Clustal Omega of SIMR-1, HPO-40, their related nematode orthologs, and the eight most significant hits from HHpred server (see Methods). The four aromatic residues that constitute the aromatic cage are highlighted in blue and the absolutely conserved arginine and aspartate residues characteristic of extended Tudor domains are highlighted in red. The location of the *simr-1[R159C]* mutation is marked with an asterisk. Cel - *C. elegans*, Cre – *C. remanei*, Cbn – *C. brenneri*, Cja – *C. japonica*, Dme – *D. melanogaster*, Hsa – *H. sapiens*, Mmu – *M. musculus*, and Bmo – *B. mori*. The online version of this article includes the following figure supplement(s) for figure 1:

**Figure supplement 1.** Identification and localization of MUT-16-associated proteins.

and MATH-33 is a ubiquitin C-terminal hydrolase that was previously identified in a proteomics screen of RDE-10-interacting proteins and RNAi screen for genes involved in co-suppression, a phenomenon where repetitive transgenes silence homologous endogenous genes (*Zhang et al., 2012*; *Robert et al., 2005*). Therefore, in total, our mass spectrometry screen identified eight proteins not previously known to be members of the *mutator* complex, five of which have no known link to any small RNA pathway.

## Localization of MUT-16-associated proteins

To determine whether any of the candidate MUT-16-associated proteins have localization patterns similar to MUT-16, we tagged each protein at its endogenous locus with a C-terminal mCherry and 2xHA tag using CRISPR. Two of the uncharacterized proteins, MATH-33 and Y57G11C.3 localize to the nucleus of germ cells and three more, F37C4.5, HGO-1, and C33G3.6, showed no obvious fluorescence in the cytoplasm or nucleus of germ cells above background levels (*Figure 1—figure supplement 1B*). In contrast, C06A5.6, which we subsequently named SIMR-1, formed distinct perinuclear foci in germ cells, either adjacent to or colocalizing with *Mutator* foci (*Figure 1B*). Similarly, RSD-2 also localized to similar perinuclear foci, in contrast to previous reports that it localizes to germ cell nuclei or the nucleolus (*Figure 1—figure supplement 1C*; *Sakaguchi et al., 2014*; *Han et al., 2008*).

Because we could not initially identify any conserved domains in SIMR-1 that would help to predict its function, we first investigated whether there are similar proteins in *C. elegans* or other related nematode species. Using BLAST, we identified a single paralog in *C. elegans*, HPO-40, and a single ortholog of both SIMR-1 and HPO-40 in *C. brenneri*, *C. remanei*, and *C. japonica*. SIMR-1 and HPO-40 are more closely related to one another than to *C. brenneri*, *C. remanei*, or *C. japonica* paralogs, suggesting that they may be a recent duplication (*Figure 1C*). We proceeded to tag HPO-40 with a C-terminal mCherry and 2xHA tag using CRISPR, and like SIMR-1, HPO-40 formed perinuclear foci in germ cells, either adjacent to or colocalizing with *Mutator* foci (*Figure 1—figure supplement 1D*).

MUT-16 is required for the localization of all known *mutator* complex proteins to *Mutator* foci (*Phillips et al., 2012*; *Uebel et al., 2018*). To determine if MUT-16 is required for SIMR-1 localization, we crossed a *mut-16* null allele into the SIMR-1::mCherry strain. Interestingly, SIMR-1 foci were still present in the *mut-16* mutant (*Figure 1D*). To address the reciprocal question, whether SIMR-1 or it's paralog HPO-40 is required for MUT-16 localization, we generated deletion alleles of both *simr-1* and *hpo-40* using CRISPR. MUT-16 foci were unperturbed in the *simr-1; hpo-40* double mutant (*Figure 1D*). These data indicate that while SIMR-1 forms germline foci near *Mutator* foci, it neither requires *Mutator* foci for its localization, nor is the localization of *Mutator* foci dependent on SIMR-1 or HPO-40, suggesting it may form separate and distinct germline foci.

## SIMR-1 contains an extended tudor domain

Interestingly, while a search of the Conserved Domain Database for either SIMR-1 or HPO-40 does not identify any conserved domains, a similar search with *C. remanei* CRE08315 weakly identifies a Tudor domain near the N-terminus (E-value 1.58e-03) (*Marchler-Bauer et al., 2011*). We next searched SIMR-1 and related protein sequences using the HHpred server, which is more sensitive than BLAST in finding remote homologs (*Söding et al., 2005*). HHpred identified homology to multiple Tudor domain-containing proteins, specifically those containing extended Tudor domains, including *D. melanogaster* Tudor, Papi and Tudor-SN, *M. musculus* TDRD1, *H. sapiens* TDRD1, TDRKH, and TDRD11, and *B. mori* Papi (*Figure 1E*). Many of these hits are Tudor domain proteins with known roles in the piRNA pathway, (*Liu et al., 2010a*; *Mathioudakis et al., 2012*; *Friberg et al., 2009*; *Ren et al., 2014*; *Zhang et al., 2017*; *Zhang et al., 2018b*). Like canonical Tudor domains, the extended Tudor domain has four conserved aromatic residues that form an 'aromatic cage' which mediates interaction with the methylated arginine (*Liu et al., 2010a*; *Liu et al., 2010b*). SIMR-1 is missing two of these four aromatic residues, making it unclear whether it is functional to recognize a methylated substrate (*Figure 1E*). It does, however, contain the absolutely conserved arginine and aspartic acid residues, which play a structural role in the extended Tudor domain (*Liu et al., 2010a*). Thus, SIMR-1 is an extended Tudor domain protein with homology to several Piwi-binding proteins. However, further experiments will be needed to determine whether it is functionally able to recognize methylated substrates.

## RNA-silencing phenotypes of MUT-16-associated proteins

If any of the previously uncharacterized proteins identified in the MUT-16 IP-mass spectrometry experiment play a role in RNA silencing, we would expect them to have phenotypes associated with siRNA-mediated gene silencing. We obtained deletion alleles in F37C4.5, *hgo-1*, and *math-33* from the *Caenorhabditis* Genetics Center (CGC) and the National Bioresource Project of Japan, and

generated new deletion alleles in *simr-1*, C33G3.6, and Y57G11C.3 by CRISPR. Strains containing mutations in *mut-16*, other known *mutator* complex proteins such as *rde-8* or *nyn-1; nyn-2*, or the RNAi-related protein, *rsd-2*, are defective in both somatic and germline exogenous RNAi

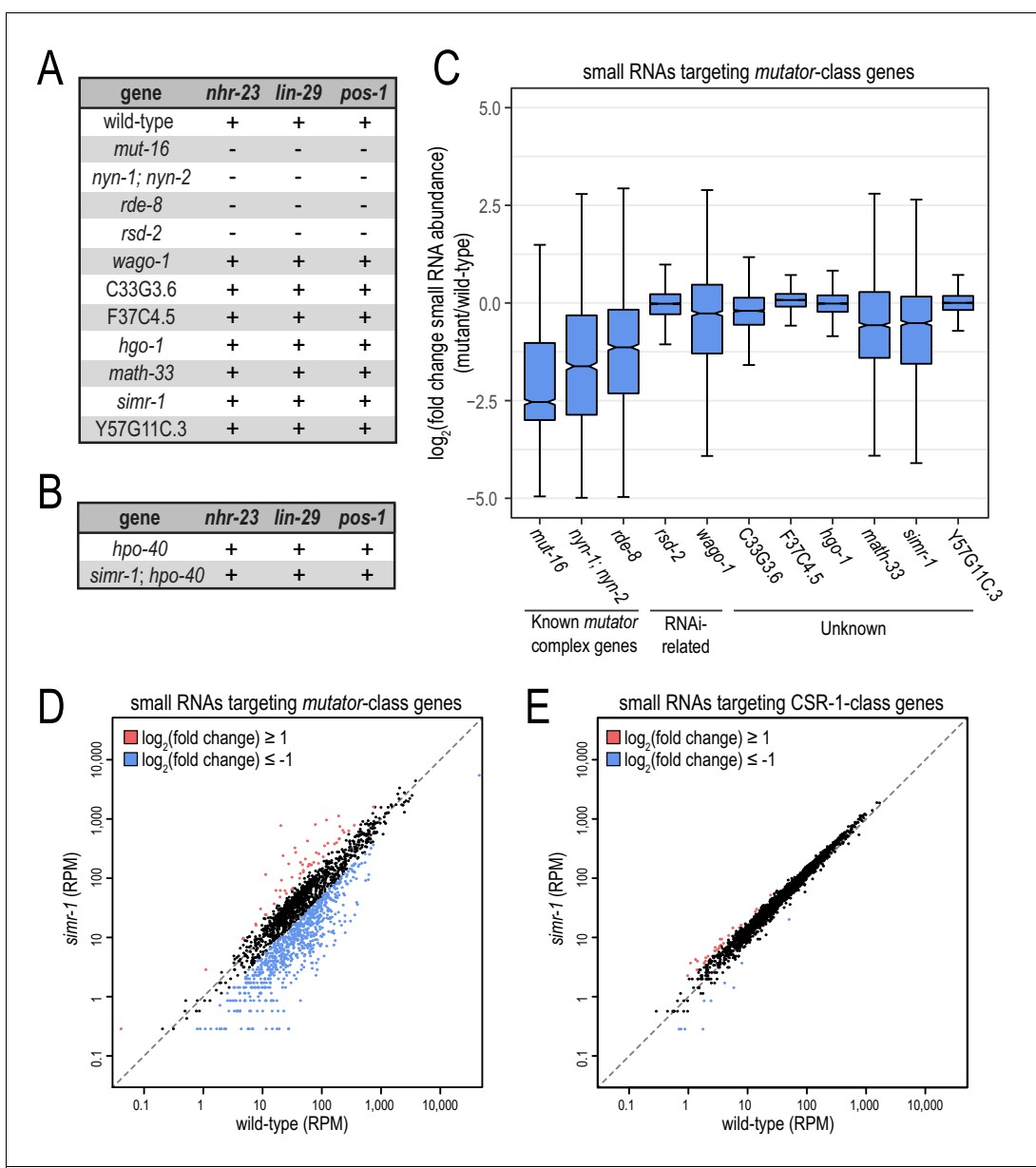

**Figure 2.** Small RNA-related phenotypes associated with deletions in MUT-16-associated proteins. (**A**) Animals carrying deletions for each previously-uncharacterized gene identified in the MUT-16 IP-mass spec experiment were assayed for their ability to respond to somatic (*nhr-23* or *lin-29*) or germline (*pos-1*) RNAi. "+" indicates wild-type response and "-" indicates RNAi-defective response. (**B**) Worms carrying deletions for *hpo-40* single mutants or *simr-1; hpo-40* double mutants were assayed for their ability to respond to somatic (*nhr-23* or *lin-29*) or germline (*pos-1*) RNAi as described in (**A**). (**C**) Box plot displaying total small RNA levels targeting *mutator*-target genes in the indicated mutant strains relative to wild-type animals. (**D,E**) Scatter plots display small RNA reads per million total reads mapping to *mutator*-target genes (**D**) and CSR-1-class genes (**E**) in wild-type and *simr-1* mutants. Genes for which log$_2$(fold change small RNA abundance)≥1 are colored dark red and genes for which log$_2$(fold change small RNA abundance)≤−1 are colored light blue.

The online version of this article includes the following figure supplement(s) for figure 2:

**Figure supplement 1.** *Mutator*-class small RNAs are reduced in *simr-1* but not *hpo-40* mutants.

(*Figure 2A*; *Zhang et al., 2012*; *Tsai et al., 2015*; *Sakaguchi et al., 2014*; *Han et al., 2008*; *Tijsterman et al., 2004*). To determine whether any of the MUT-16-associated proteins play a role in exogenous RNAi, we tested the deletion alleles on both somatic and germline RNAi. All deletions, including *simr-1*, elicited RNAi phenotypes similar to wild-type animals indicating that these genes are not required for exogenous RNAi (*Figure 2A*). We hypothesized that *simr-1* could be redundant with its paralog, *hpo-40*, so we additionally tested *hpo-40* single mutants and *simr-1; hpo-40* double mutants. Both the single and double mutants elicited RNAi phenotypes similar to wild-type animals indicating that neither *hpo-40* alone nor the two proteins acting together are required for exogenous RNAi (*Figure 2B*).

To assess the levels of endogenous siRNAs in each deletion mutant, we isolated RNA from synchronous 1 day adult animals and generated small RNA sequencing libraries. Because these proteins were identified by MUT-16 IP-mass spec, we focused on a group of approximately 2000 genes that are known targets of the *mutator* pathway (*Lee et al., 2012*; *Gu et al., 2009*; *Phillips et al., 2014*; *Zhang et al., 2011*; *Tsai et al., 2015*). We observed a substantial reduction in total small RNAs mapping to these *mutator*-target genes when known components of the *mutator* complex or RNA silencing pathway, such as *mut-16*, *wago-1*, *rde-8*, or *nyn-1; nyn-2* are disrupted (*Figure 2C*). We also observed a reduction in small RNAs mapping to the *mutator*-target genes, albeit more modest, in *math-33* and *simr-1* mutants (*Figure 2C–D*). However, due to asynchrony and slow growth of the *math-33* mutant animals that could confound the data analysis, we chose not to further analyze the libraries made from this strain at this time. In contrast to the *mutator*-target genes, we observed no change in total small RNAs mapping to CSR-1-target genes in the *simr-1* mutant (*Figure 2E*). To test for redundancy between *simr-1* and its paralog, *hpo-40,* in the endogenous siRNA pathway, we additionally examined levels of small RNAs mapping to *mutator*-target genes in *hpo-40* single mutants and *simr-1; hpo-40* double mutants. We observed no significant reduction in *mutator*-dependent small RNAs in the *hpo-40* single mutant, and the reduction in *mutator*-dependent small RNAs in the *simr-1; hpo-40* double mutant resembled that of the *simr-1* single mutant (*Figure 2—figure supplement 1A–B*). Therefore, we concluded that SIMR-1 alone is required for siRNA production at some *mutator*-target genes.

## *simr-1* mutants have a mortal germline at elevated temperature

Mutations in the *mutator* pathway are temperature-sensitive sterile, while mutations in other related small RNA pathways have a variety of fertility defects (*Ketting et al., 1999*; *Zhang et al., 2011*). For example, mutations in the *C. elegans* ortholog of Piwi, *prg-1*, which associates with piRNAs, display a progressive sterility that accumulates over many generations (also referred to as a Mortal Germline or Mrt phenotype), and mutations in nuclear RNAi pathway genes *nrde-1*, *nrde-2*, *nrde-4*, and *hrde-1* or in the *rsd-2* and *rsd-6* genes elicit a similar Mrt phenotype, but only at elevated temperature (*Simon et al., 2014*; *Sakaguchi et al., 2014*; *Buckley et al., 2012*). In order to determine if *simr-1* mutants have fertility defects or the Mrt phenotype observed in many other small RNA silencing pathway mutants, we quantified their brood size at 20°C, and after every generation at 25°C for 11 generations. *mut-16* mutants were included as a control and, as expected, fertility was reduced by 95.3% in the first generation at 25°C, with the few fertile animals producing only sterile progeny by the second generation at 25°C (*Figure 3A*). In contrast, wild-type animals displayed a 40.3% reduction in brood size and *simr-1* mutants displayed a 59.0% reduction in brood size after a single generation at 25°C compared to 20°C (*Figure 3A*). However, unlike wild-type animals which remained fertile after more than 11 generations at 25°C, *simr-1* mutants became progressively sterile over the next 10 generations at 25°C until reaching complete sterility at generation 11 (*Figure 3A*). We additionally tested the fertility of the *hpo-40* single mutant, which was indistinguishable from wild-type, and the *simr-1; hpo-40* double mutant which became sterile after approximately 11 generations, similar to the *simr-1* single mutant (*Figure 3—figure supplement 1*). These data indicate that loss of *simr-1* at elevated temperature triggers a molecular defect that is cumulative and ultimately results in loss of fertility.

Because small RNA pathways play key roles in the regulation of transposons, one hypothesis would be that increased DNA mutations triggered by transposon mobilization in *simr-1* mutants at 25°C lead to reduced fertility over the course of multiple generations. To address this possibility, we selected wild-type and *simr-1* mutant animals raised for 10 generations at 25°C, and returned them to 20°C. Within approximately four generations at 20°C, the fertility of *simr-1* mutants recovered to

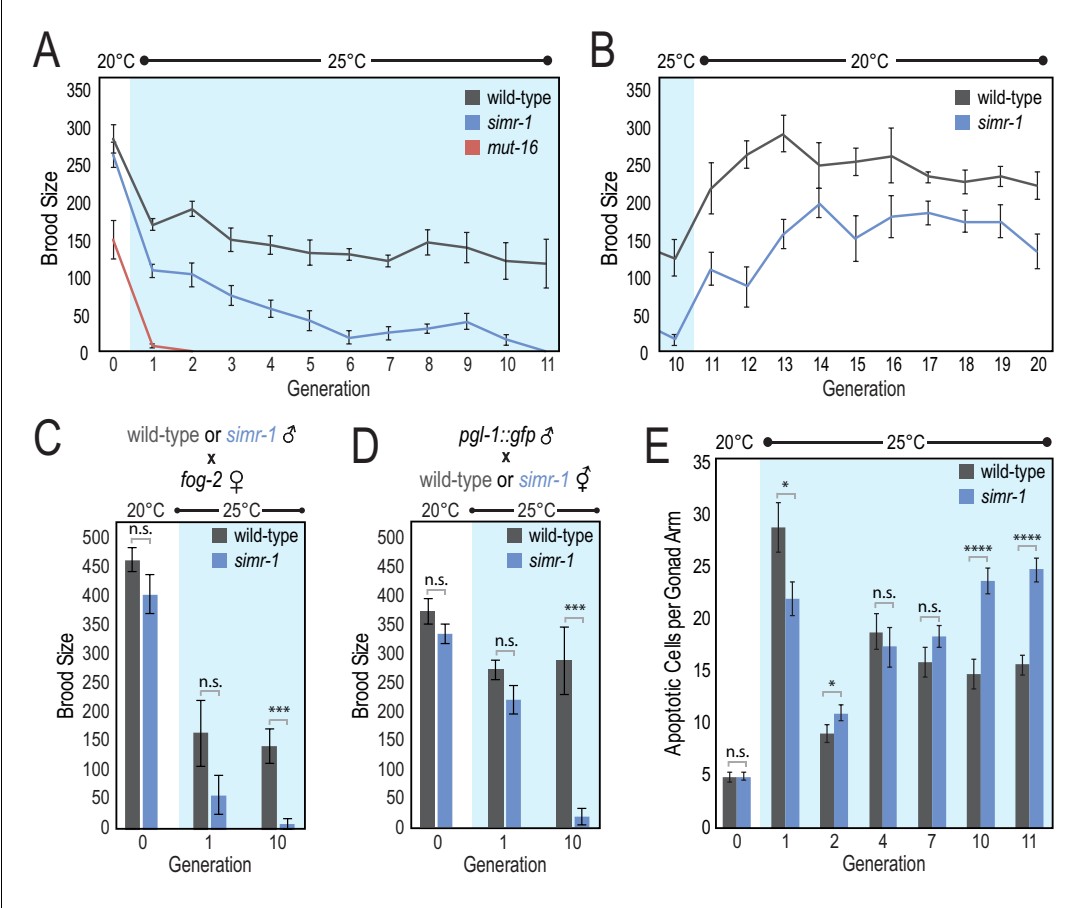

**Figure 3.** *simr-1* mutants have a transgenerational fertility defect at elevated temperature. (**A**) Brood size was scored for a single generation at 20°C, followed by 11 generations at 25°C, demonstrating that *simr-1* mutants become progressively sterile at 25°C. 10 broods were scored for each genotype at each generation. (**B**) Brood sizes for *simr-1* mutant and wild-type animals were scored for 10 generations after returning to 20°C, following 10 generations at 25°C, demonstrating restoration of fertility at permissive temperature. 10 broods were scored for each genotype at each generation. (**C**) Wild-type and *simr-1* mutant males were raised either at 20°C, a single generation at 25°C, or following 10 generations of growth at 25°C, and then mated to *fog-2* females raised at 20°C. Brood sizes were scored for 10 *fog-2* females, each mated to four males of the indicated genotypes, and demonstrating that *simr-1* male fertility is compromised at 25°C. (**D**) Wild-type and *simr-1* mutant hermaphrodites were raised either at 20°C, a single generation at 25°C, or following 10 generations of growth at 25°C, and then mated to four *pgl-1::gfp* males raised at 20°C. Brood sizes were scored for each of 10 wild-type or *simr-1* mutant hermaphrodites, mated to four *pgl-1::gfp* males. Only plates with GFP positive progeny were scored. These data indicate that oogenesis of *simr-1* is compromised after multiple generations at 25°C. (**E**) Number of apoptotic germ cells were counted in a minimum of 20 wild-type and *simr-1* mutant gonads using CED-1::GFP engulfment as a marker for apoptotic germ cells. Animals were raised either at 20°C, or for one, two, four, seven, 10 or 11 generations at 25°C, and imaged approximately 24 hr after the L4 larval stage. Error bars indicate SEM. n.s. denotes not significant and indicates a p-value>0.05, * indicates a p-value≤0.05, *** indicates a p-value≤0.001, **** indicates a p-value≤0.0001. See *Supplementary file 8* for more details regarding statistical analysis.

The online version of this article includes the following source data and figure supplement(s) for figure 3:

**Source data 1.** Data used to generate *Figure 3A* and *Figure 3—figure supplement 1*.
**Source data 2.** Data used to generate *Figure 3B*.
**Source data 3.** Data used to generate *Figure 3C*.
**Source data 4.** Data used to generate *Figure 3D*.
**Source data 5.** Data used to generate *Figure 3E*.
**Figure supplement 1.** *hpo-40* does not contribute to the progressive sterility of *simr-1* mutants.

within 72.8% of pre-25°C levels (*Figure 3B*). These data indicate that the reduction in *simr-1* fertility at 25°C is not primarily due to the accumulation of DNA mutations, but may be due to transcriptional or chromatin changes that can be reset after recovery at 20°C, similar to what has been observed previously for *hrde-1* and *hrde-2* (*Spracklin et al., 2017*; *Ni et al., 2016*).

## *simr-1* Mrt phenotype results from defective sperm and oocytes

To determine whether the Mrt phenotype observed in *simr-1* mutants at 25°C is due to defects in oogenesis or spermatogenesis we conducted mating assays. First, we crossed wild-type or *simr-1* mutant males raised at 20°C, a single generation at 25°C, or after 10 generations at 25°C to *fog-2* females, which cannot make their own sperm, raised at 20°C. *simr-1* mutant males raised for a single generation at 25°C sired fewer progeny than the wild-type control males, and *simr-1* mutant males raised for 10 generations at 25°C were nearly sterile, similar to *simr-1* hermaphrodites raised for 10 generations at 25°C (*Figure 3C*). We next sought to address whether *simr-1* mutant oocytes are similarly compromised. Males expressing fluorescently tagged *pgl-1::gfp* (*Andraljoc et al., 2017*), were mated to *simr-1* mutant hermaphrodites raised at 20°C, a single generation at 25°C, or after 10 generations at 25°C. The *pgl-1::gfp* males were used to easily distinguish between cross progeny and self progeny from the *simr-1* mutant hermaphrodites. *simr-1* mutant hermaphrodites raised for a single generation at 25°C and provided with wild-type sperm produced a similar number of progeny to a wild-type control. In contrast, after 10 generations at 25°C, *simr-1* mutant hermaphrodites were nearly sterile, even when provided with wild-type sperm (*Figure 3D*). These data indicate that both spermatogenesis and oogenesis are defective in *simr-1* mutants raised at elevated temperature for multiple generations.

## *simr-1* Mrt phenotype is associated with increased levels of germ cell apoptosis

Apoptosis occurs in the late pachytene region of the germline where approximately half of all germ cells are eliminated by physiological apoptosis in a wild-type animal (*Gumienny et al., 1999*). DNA damage or other stressful conditions can trigger an increase in apoptosis as part of a quality control mechanism (*Gartner et al., 2000*; *Gartner et al., 2008*). To determine if *simr-1* mutant gonads have increased apoptosis, we introduced the CED-1::GFP reporter, which allows visualization of apoptotic germ cells, into the *simr-1* mutant (*Schumacher et al., 2005*). We observed no significant differences in apoptotic germ cells at 20°C (*Figure 3E*). After a single generation at 25°C, we observe a dramatic increase in apoptotic germ cells, with apoptosis levels modestly higher in wild-type compared to *simr-1* mutants. This spike in apoptotic germ cells in the first generation at 25°C is followed by a reduction in apoptosis in the second generation at 25°C. However, only after 10 or 11 generations at 25°C does the number of apoptotic germ cells in *simr-1* mutants rise significantly compared to wild-type animals (*Figure 3E*). These data suggest that an increase in germ cell dysfunction in *simr-1* mutant animals after multiple generations of growth at 25°C is associated with both increased germ cell apoptosis and reduced fertility. Nonetheless, it is important to note that similar levels of apoptotic germ cells are observed in fertile wild-type animals after only one generation at 25°C, indicating that a high level of apoptosis is not always directly correlative with sterility.

## Mutations in *simr-1* desilence a piRNA sensor but not an ERGO-1-dependent siRNA sensor

In a previously described mutagenesis screen, we identified novel genes acting in the piRNA-mediated silencing pathway using a strain expressing GFP::H2B carrying a piRNA target in its 3'UTR (the 'piRNA sensor') (*Bagijn et al., 2012*; *de Albuquerque et al., 2014*). Because the piRNA sensor is subject to siRNA-mediated heritable silencing (RNAe) making it no longer susceptible to desilencing when the piRNA pathway is compromised, the screen was performed in a *henn-1* mutant background, which partially desilences this transgene and allows for the identification of both piRNA pathway and secondary siRNA pathway mutants (*Kamminga et al., 2012*). From this screen we identified two alleles of *simr-1* that further desilence the piRNA sensor transgene in the *henn-1* mutant background (*Figure 4A*). The first, *simr-1[A11V]*, is found in a well-conserved region near the N-terminus of the protein and the second, *simr-1[R159C]*, is the absolutely conserved arginine that plays a structural role in the extended Tudor domain (*Figure 1E*). Interestingly, when we crossed our *simr-1* deletion mutant into the piRNA sensor strain without the *henn-1* mutant, we observed that *simr-1* was not sufficient to desilence the piRNA sensor transgene in the absence of the *henn-1* mutant (*Figure 4—figure supplement 1A*), similar to what has been observed previously with *prg-1* (*Luteijn et al., 2012*). In contrast, a mutation in *mut-16* robustly desilences the same piRNA sensor transgene (*Figure 4—figure supplement 1A*). These data indicate that a mutation in *simr-1*, like

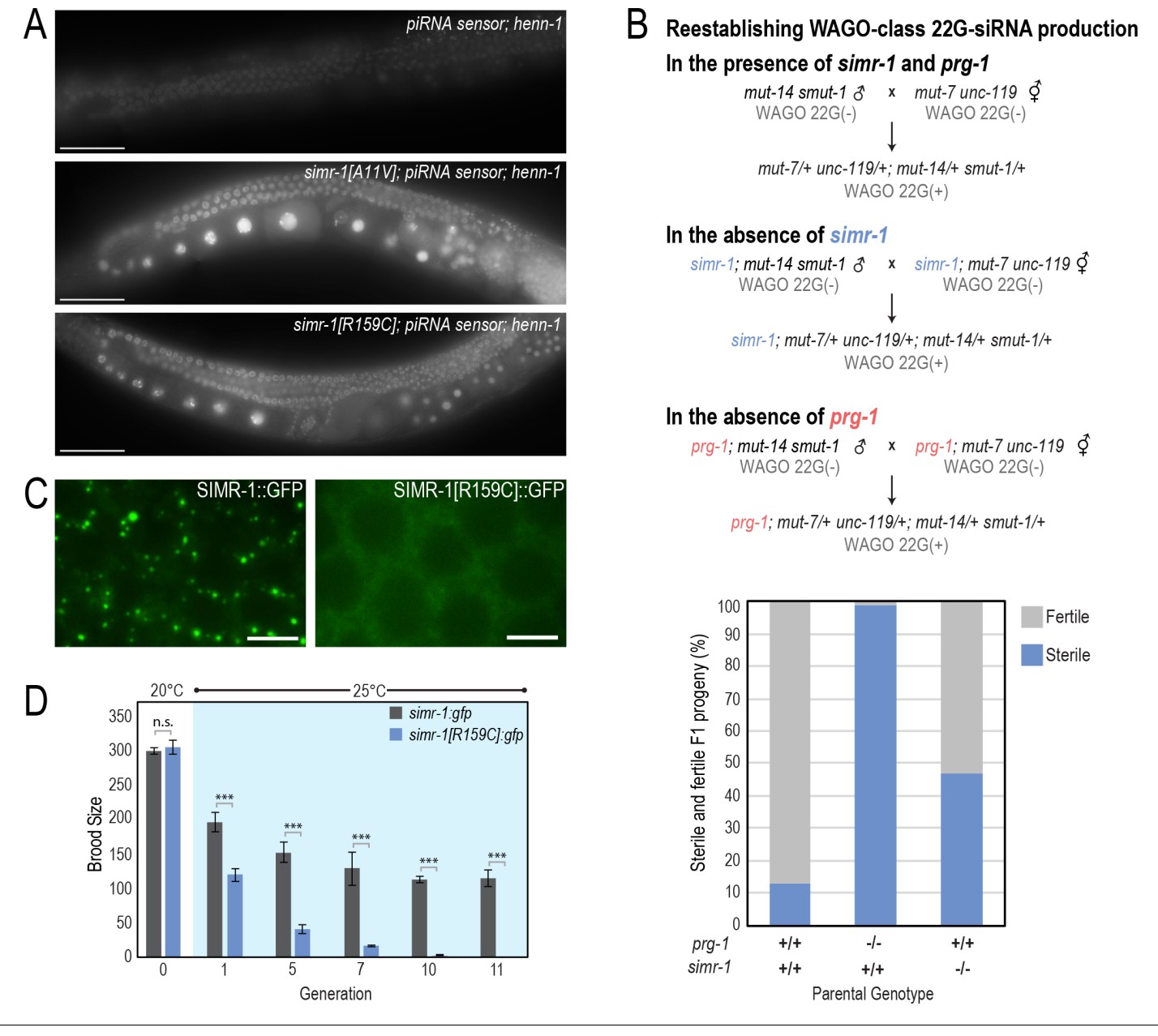

**Figure 4.** *simr-1* mutants have piRNA-related defects. (**A**) Images of adult animals, in which the *henn-1* mutation weakly desilences the piRNA sensor (top). *simr-1[A11V]* (middle) and *simr-1[R159C]* (bottom) mutants, obtained from an EMS mutagenesis screen of the *henn-1; piRNA* sensor strain further desilence the sensor and increase GFP expression. All images were obtained using the same microscope settings. Scale bars, 50 μm. (**B**) A mating-based approach to reestablish WAGO-class 22G-siRNA production in the presence and absence of *simr-1* and *prg-1*. Schematic (top) illustrating the three crosses and bar graph (bottom) showing percentage of fertile and sterile animals from each cross. (**C**) Live imaging of SIMR-1::GFP (left) and SIMR-1[R159C]::GFP (right) demonstrate that Tudor domain is critical for SIMR-1 localization to perinuclear foci. Scale bars, 5 μm. (**D**) Brood size was scored for *simr-1::gfp* and *simr-1[R159C]::gfp* strains at 20˚C, then animals were raised for 11 generations at 25˚C. Broods were additionally scored at generations one, five, seven, 10 and 11 at 25˚C demonstrating that the *simr-1[R159C]::gfp* strain becomes progressively sterile at 25˚C, similar to the *simr-1* null mutation, while *simr-1::gfp* maintains fertility at 25˚C similar to wild-type animals. 10 broods were scored for each genotype at each generation.

The online version of this article includes the following source data and figure supplement(s) for figure 4:

**Source data 1.** Data used to generate *Figure 4B*.

**Source data 2.** Data used to generate *Figure 4D*.

**Figure supplement 1.** *simr-1* mutants do not display defects associated with mutants in the *mutator* or ERGO-1 26G-siRNA pathways.

*prg-1*, is sufficient to desilence a sensitized piRNA sensor strain, but cannot reactivate a piRNA sensor silenced by RNAe.

To examine the role of SIMR-1 in other small RNA pathways, we next introduced a *simr-1* mutant into the 22G-siR1 sensor which is sensitive to perturbations in the ERGO-1 26G-siRNA pathway and the downstream *mutator* pathway (*Montgomery et al., 2012*). A mutation in *simr-1* was unable to desilence the 22G-siR1 sensor (*Figure 4—figure supplement 1B*). In contrast, a mutation in *mut-16* robustly desilenced the 22G-siR1 sensor (*Figure 4—figure supplement 1B*). Furthermore, when animals with mutations in the ERGO-1 26G-siRNA pathway, like *eri-7* (*Fischer et al., 2008*), are fed *lir-1*, *hmr-1*, or *dpy-13* double-strand RNA, they display an Enhanced RNAi (Eri) phenotype which was not observed with the *simr-1* mutant (*Figure 4—figure supplement 1C*). These data indicate that SIMR-1 is not required for silencing of genes targeted by the ERGO-1 26G-siRNA pathway.

## SIMR-1 is required to prevent sterility after reestablishing WAGO-class 22G-siRNA production

Neither the *mutator* pathway nor the piRNA pathway are essential for fertility under normal growth conditions (*Ketting et al., 1999*; *Zhang et al., 2011*; *Batista et al., 2008*; *Wang and Reinke, 2008*; *Simon et al., 2014*). Nonetheless, restoration of the *mutator* pathway, and therefore RNA silencing by WAGO-class 22G-siRNAs, in a strain lacking both the *mutator* pathway and the piRNA pathway, causes sterility (*de Albuquerque et al., 2015*; *Phillips et al., 2015*). This sterility is a direct result of the routing of essential genes into the *mutator* pathway and indicates that inheritance of piRNAs from one generation to the next is critical to ensuring that the correct genes are silenced by the *mutator* pathway. To determine whether *simr-1*, like *prg-1*, is required to maintain fertility when resetting the *mutator* pathway, we crossed two strains to one another containing distinct mutations in the *mutator* pathway, *mut-7* and *mut-14 smut-1*, such that their progeny would inherit a wild-type copy of *mut-7* from one parent, a wild-type copy *mut-14 smut-1* from the other, and thus would be competent to produce WAGO-class 22G-siRNAs (*Figure 4B*). The hermaphrodite strain always additionally carried the *unc-119* mutation, which allowed us to easily distinguish between self progeny which have the Uncoordinated (Unc) phenotype and cross progeny which have wild-type movement. If *simr-1* is required for the proper functioning of the piRNA pathway, we would predict that when it, like *prg-1*, is introduced into the two strains used to reset the *mutator* pathway the progeny of the cross will be sterile. In fact, this result is what we observed. In the control cross (*mut-14 smut-1* males mated to *mut-7 unc-119* hermaphrodites), only 13.0% of the F1 heterozygous progeny were sterile (*Figure 4B*). In contrast, when the *simr-1* mutation is present in both parental strains (*simr-1; mut-14 smut-1* males mated to *simr-1; mut-7 unc-119* hermaphrodites) the percentage of sterile progeny increased to 47.1%, and for the *prg-1* cross (*prg-1; mut-14 smut-1* males mated to *prg-1; mut-7 unc-119* hermaphrodites), the number of sterile animals increases further to 98.8% (*Figure 4B*). These results indicate that *simr-1*, like *prg-1*, is required during establishment of the *mutator* pathway to promote fertility, likely by directing *mutator*-dependent silencing to piRNA-targeted genes.

## The tudor domain of SIMR-1 is required for its localization and function

To determine whether the Tudor domain of SIMR-1 is necessary for its localization to germline foci, we used CRISPR to engineer the R159C mutation into the *simr-1::gfp* strain. The R159C allele, isolated from a mutagenesis of the *henn-1;* piRNA sensor strain, is predicted to disrupt the conformation of the extended Tudor domain (*Liu et al., 2010a*). By live imaging, we observed that SIMR-1 [R159C]::GFP no longer forms germline foci, despite its clear expression in the cytoplasm of germ cells (*Figure 4C*). We further confirmed that SIMR-1[R159C]::GFP is expressed at wild-type levels by western blot (*Figure 4—figure supplement 1D*). These data indicate that an intact extended Tudor domain is not required for SIMR-1 expression but is essential for the localization of SIMR-1 to germline foci.

We next investigated whether the *simr-1[R159C]::gfp* strain exhibited fertility defects at elevated temperature. Like the *simr-1* deletion allele, *simr-1[R159C]::gfp* exhibited progressive sterility at elevated temperature, becoming sterile after approximately 10–11 generations (*Figure 4D*). In contrast, the wild-type *simr-1::gfp* remained fertile for the duration of the experiment (*Figure 4D*). Together, these data show that the extended Tudor domain is essential for SIMR-1 function, and that

disruption of the Tudor domain results in loss of SIMR-1 germline foci and causes a Mrt phenotype similar to that of the *simr-1* deletion allele.

## SIMR-1 is required for small RNA production at piRNA-target genes

To comprehensively characterize the role of SIMR-1 in *C. elegans* endogenous small RNA pathways, we generated small RNA libraries from wild-type and *simr-1* mutants at 20℃ and after culturing for one, two, seven, or 10 generations at 25℃. For comparison, we also generated small RNA libraries from wild-type, *mut-16,* and *prg-1* mutants at 20℃ and from wild-type and *mut-16* mutants cultured for a single generation at 25℃. In *simr-1* mutants, 817 genes were depleted of small RNAs and 213 genes were enriched for small RNAs at 20℃ when compared to wild-type at 20℃ (*Figure 5A* and *Supplementary file 3*). After one generation at 25℃, 1258 genes were depleted of small RNAs and 2712 genes were enriched for small RNAs compared to wild-type also cultured for one generation at 25℃ (*Figure 5A* and *Supplementary file 3*). When *simr-1* mutants were then cultured for two, seven, or 10 generations at 25℃, 927, 885, and 907 genes were depleted of small RNAs and 194, 110, and 100 genes were enriched for small RNAs, respectively, when compared to both wild-type cultured at 25℃ for one generation and wild-type cultured at 25℃ in parallel to *simr-1* for an equal number of generations (*Figure 5A* and *Supplementary file 3*). These data implicate SIMR-1 in the production or maintenance of small RNAs at many *C. elegans* genes.

siRNAs can be classified based on their Argonaute protein binding partner and the other proteins or protein complexes required for their biogenesis. To identify the small RNA pathway(s) in which SIMR-1 plays a role, we looked at the change in total small RNA levels at groups of genes known to be targets of the CSR-1, *mutator*, piRNA, or ERGO-1 pathways in *simr-1* mutants compared to wild-type at both 20℃ and a single generation at 25℃ (*Lee et al., 2012*; *Fischer et al., 2011*; *Gu et al., 2009*; *Phillips et al., 2014*; *Zhang et al., 2011*; *Tsai et al., 2015*). Small RNAs derived from CSR-1-target genes were modestly up-regulated at 20℃ and more dramatically up-regulated after a single generation 25℃ in *simr-1* mutants (*Figure 5B–C* and *Figure 5—figure supplement 1A*). In contrast, small RNAs from *mutator*-target genes and piRNA-target genes were reduced in *simr-1* mutants at both 20℃ and 25℃ (*Figure 5B–C* and *Figure 5—figure supplement 1A*). piRNA target genes make up the majority of *mutator*-target genes (*Figure 5D*). To determine if piRNA-target genes are more severely reduced of small RNAs in *simr-1* mutants than other *mutator*-target genes, we generated a list of *mutator*-target genes whose small RNAs are either unchanged or increased in *prg-1* mutants (log$_2$(fold change small RNA abundance)≥0 in *prg-1* mutants relative to wild-type). These PRG-1-independent *mutator*-target genes are not reduced of small RNAs compared to all siRNA target genes and are significantly less depleted of small RNAs compared to all *mutator*-target genes or piRNA-target genes (*Figure 5B*). Furthermore, the well-characterized endogenous RDE-1 target, Y47H10A.5 (*Corrêa et al., 2010*), was not depleted of small RNAs in *simr-1* mutants at either 20℃ or 25℃ or in *prg-1* mutants at 20℃, but was severely depleted of small RNAs in *mut-16* mutants at both 20℃ and 25℃ (*Figure 5—figure supplement 1B*), demonstrating that like exogenous RNAi targets (*Figure 2A*), small RNA levels at endogenous RDE-1 targets are not affected in the *simr-1* mutant. Small RNAs from ERGO-1 target genes were reduced mildly at 20℃ and more severely at 25℃ (*Figure 5B* and *Figure 5—figure supplement 1A*), however because *simr-1* was unable to desilence the 22G-siRNA sensor and did not have an Eri phenotype (*Figure 4—figure supplement 1B–C*), we did not pursue further investigation of the ERGO-1 pathway. Therefore, these data indicate that SIMR-1 is important for the production of high levels of endogenous small RNAs at many *mutator*-target genes, including primarily piRNA-target genes, but is not required for small RNA production at CSR-1-target genes or at endogenous and exogenous RDE-1-target genes.

## SIMR-1 is not required for piRNA biogenesis or stability

84% of genes with reduced small RNAs in a *simr-1* mutant at 20℃ also have reduced small RNAs in a *prg-1* mutant at 20℃ (*Figure 5D*). This reduction of siRNAs at piRNA-target genes could result from a loss of piRNAs in the *simr-1* mutant animals, or alternatively, piRNAs could be expressed at wild-type levels and only the downstream siRNAs could be affected. To address these possibilities, we counted the number of reads mapping to annotated piRNA loci in wild-type and *simr-1* mutants. Similarly to what has been previously reported, piRNA expression is significantly reduced at 25℃ compared to 20℃ in wild-type animals (*Belicard et al., 2018*). However, we observed no significant

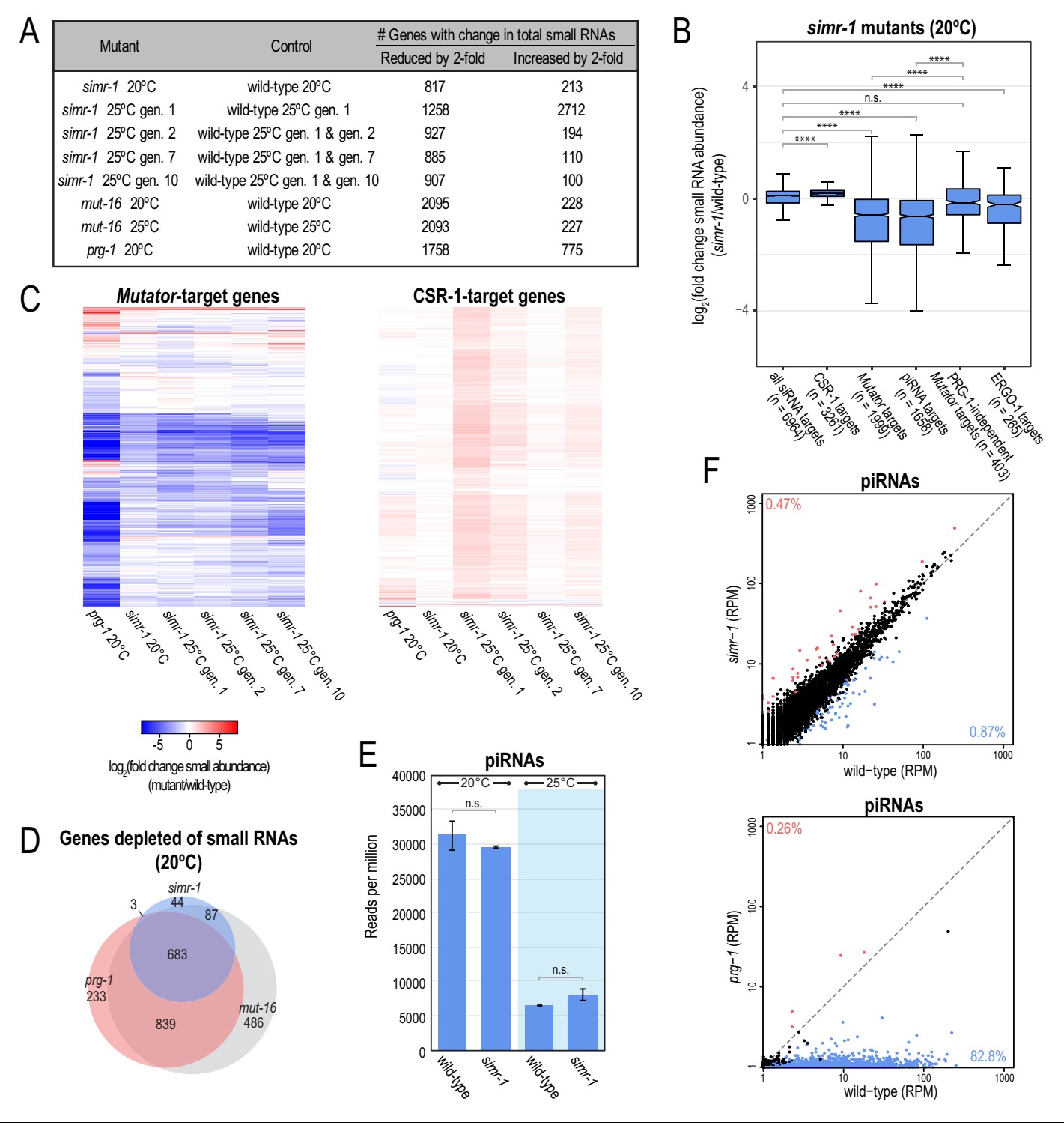

**Figure 5.** *simr-1* mutants display reduced small RNAs mapping to *mutator* and piRNA-target genes. (**A**) Table indicating the number of genes for which the total small RNA levels are either increased or reduced by at least two-fold for each indicated mutant. All genes also met the requirements of having at least 10 RPM in either mutant or control and a DESeq2 adjusted p-value of ≤0.05. (**B**) Box plots displaying total small RNAs levels mapping to genes from the indicated small RNA pathways in *simr-1* mutants compared to wild-type animals raised at 20℃. Details regarding definition of small RNA target gene classes is provided in the Materials and Methods section. At least 10 RPM in wild-type or *simr-1* mutant libraries was required to be included in the analysis. (**C**) Heat maps displaying total small RNAs levels targeting *mutator*-target genes or CSR-1-target genes in *simr-1* mutants raised at 20℃, a single generation at 25℃, or two, seven, or 10 generations at 25℃ relative to wild-type at the same temperature and generation. (**D**)
*Figure 5 continued on next page*

*Figure 5 continued*

Venn diagrams indicating overlap of genes depleted of total small RNAs by two-fold or more in mutants compared to wild-type. (E) Reads per total million reads mapping to piRNA and piRNA-target gene loci in wild-type and *simr-1* mutants raised at either 20°C, or for a single generation at 25°C, indicate that piRNAs are not reduced in *simr-1* mutants. Error bars indicate standard deviation of two replicate libraries. (F) Scatter plots display piRNA reads per million total reads in wild-type and *simr-1* mutants (top) and wild-type and *prg-1* mutants (bottom). Genes with two-fold increase in piRNA abundance and DESeq2 adjusted p-value≤0.05 are colored dark red and genes with two-fold reduction in piRNA abundance and DESeq2 adjusted p-value≤0.05 are colored light blue. The percentage of total piRNAs with an increase or reduction of greater than two-fold is indicated in the corners of the graph. n.s. denotes not significant and indicates a p-value>0.05 and **** indicates a p-value≤0.0001. See *Supplementary file 8* for more details regarding statistical analysis.

The online version of this article includes the following figure supplement(s) for figure 5:

**Figure supplement 1.** Small RNAs are reduced at many *mutator*, piRNA, and ERGO-1 target genes in *simr-1* mutants at 25°C.

difference between total piRNA levels in *simr-1* mutants compared to wild-type animals at either temperature (*Figure 5E*). We next determined whether individual piRNAs are increased or reduced in expression in *simr-1* mutants. In contrast to *prg-1* mutants in which 83% of piRNAs are reduced by at least two-fold, in *simr-1* mutants less than 1% of piRNAs are reduced by at least two-fold (*Figure 5F*). We next identified predicted piRNA target genes for the piRNAs that were reduced by at least two-fold in *simr-1* mutants (*Shen et al., 2018*; *Zhang et al., 2018a*; *Wu et al., 2018*; *Wu et al., 2019*). Specifically, we selected genes predicted to be targets for our *simr-1*-depleted piRNAs by piRTarBase using relaxed piRNA targeting rules and identified by CLASH data (*Supplementary file 4*). Of the 37 predicted target genes for our *simr-1*-depleted piRNAs, only five have reduced small RNAs in *simr-1* mutants (*Supplementary file 4*), indicating that the *simr-1*-depleted piRNAs are not a major driver of siRNA depletion in *simr-1* mutants. These data together indicate that SIMR-1 functions downstream of piRNA biogenesis.

## Small RNAs are progressively depleted across generations from some piRNA-target loci at 25°C

Because *simr-1* mutant animals become sterile after approximately 10 generations at 25°C, we next examined how the levels of small RNAs generated from *mutator* and piRNA-target genes change after two, seven, or 10 generations at 25°C, compared to a single generation at 25°C. At each generation, we compared the genes that lose small RNAs by at least two-fold in the *simr-1* mutant to genes that lose small RNAs by at least two-fold in *mut-16* mutants at 25°C, and to *prg-1* mutants at 20°C. At all generations, SIMR-1-dependent siRNA target genes largely overlapped with *mut-16*-dependent siRNA target genes. Specifically, 80%, 88%, 97% and 89% of SIMR-1-dependent small RNA target genes at 25°C for one, two, seven, and 10 generations are reduced of small RNAs in *mut-16* mutants, respectively, compared to 84% for SIMR-1-dependent siRNA target genes at 20°C (*Figure 5D* and *Figure 5—figure supplement 1C*). We next examined the overlap of SIMR-1-dependent small RNA target genes with *prg-1* mutants at 20°C. 55%, 64%, 72%, 70% of SIMR-1-dependent small RNA target genes at 25°C for one, two, seven, and 10 generations are reduced of small RNAs in *prg-1* mutants at 20°C, respectively, compared to 84% for SIMR-1-dependent siRNA target genes at 20°C (*Figure 5D* and *Figure 5—figure supplement 1C*). While the overlap of SIMR-1-dependent small RNA target genes with piRNA-dependent small RNA target genes is reduced at 25°C compared to 20°C, at least some of this difference may be attributed to the sequencing of *prg-1* mutant small RNA libraries from animals raised at 20°C only. In fact, the total number of genes reduced of small RNAs in *simr-1* mutants that overlap with piRNA-target genes remains similar between temperatures and across generations (*Figure 5D* and *Figure 5—figure supplement 1C*). However, while the number of piRNA-target genes that lose small RNAs in a *simr-1* mutant doesn't change significantly with temperature or later generations, we do observe a modest but significant progressive reduction in the number of small RNAs mapping to all piRNA-target genes corresponding to the number of generations at 25°C (*Figure 5—figure supplement 1D*). Because the number of *simr-1*-target genes does not become substantially greater after 10 generations at elevated temperature, these data indicate that the observed sterility is not due to loss of small RNAs from more loci after 10 generations. Furthermore, while many piRNA-target genes become more depleted of small RNAs after 10 generations at elevated temperature, this loss of small RNAs is unlikely to be a

contributing factor to the progressive loss of fertility in these animals because small RNA loss is even more severe in fertile *prg-1* mutants at 20°C (*Figure 5C* and S5D).

## SIMR-1 is required for small RNA production at many piRNA-targeted transposons and repetitive elements

The *mutator* pathway is required for the production of siRNAs at many transposons and repeat loci, and in the absence of *mut-16* or other *mutator* complex proteins transposon activity has been detected for at least seven distinct families of DNA transposons (Tc1-Tc5, Tc7, CemaT1) (*Eide and Anderson, 1985*; *Collins et al., 1989*; *Levitt and Emmons, 1989*; *Yuan et al., 1991*; *Collins and Anderson, 1994*; *Rezsohazy, 1997*; *Bessereau, 2006*; *Brownlie and Whyard, 2004*). In contrast, only a single transposon family, Tc3, has been demonstrated to transpose upon loss of the piRNA machinery, though several other DNA transposon loci are up-regulated at the mRNA level or lose *mutator*-dependent siRNAs (*Das et al., 2008*; *Bagijn et al., 2012*; *McMurchy et al., 2017*; *Wallis et al., 2019*; *Reed et al., 2020*). To address the role of SIMR-1 in the regulation of transposons and repeat loci, we first defined a list of *mut-16*-dependent transposons and repeats using a cutoff of two-fold reduction of small RNAs in the *mut-16* mutant compared to wild-type at 20°C. All features also met the requirements of having at least 10 RPM in either mutant or wild-type and a DESeq2 adjusted p-value of ≤0.05. Of these *mut-16*-dependent transposons and repeats, 11% and 25% of transposons at 20°C and 25°C respectively, and 35% and 45% of repeat loci, at 20°C and 25°C respectively, were reduced by two-fold or greater of small RNAs in *simr-1* mutants compared to wild-type (*Figure 6—figure supplement 1A*). Furthermore, 82% of the *mut-16*-dependent transposons depleted of small RNAs by greater than two-fold in *simr-1* mutants at 20°C were also depleted in *prg-1* mutants at 20°C (*Figure 6—figure supplement 1B*). Similarly, 80% of the *mut-16*-dependent repeats depleted of small RNAs by greater than two-fold in *simr-1* mutants at 20°C were also depleted in *prg-1* mutants at 20°C (*Figure 6—figure supplement 1B*). We next focused on transposons for which silencing is known to be either piRNA-dependent or piRNA-independent. Transposon Tc3 becomes active in mutants of the *mutator* pathway and the piRNA pathway, while Tc1 and Tc4 activity is specific to the *mutator* pathway (*Das et al., 2008*). Tc2 activity has not been measured in piRNA pathway mutants, but the Tc2 transposase mRNA is significantly up-regulated in a *prg-1* mutant (*Wallis et al., 2019*). We next determined the number of small RNAs mapping to these four transposon sequences in *simr-1* mutants compared to wild-type. Small RNAs mapping to Tc2 and Tc3 were significantly reduced in both the *simr-1* mutant as well as in a *mut-16* mutant, at both 20°C and 25°C (*Figure 6A* and *Figure 6—figure supplement 1C*). In contrast, small RNAs mapping to Tc1 and Tc4v, the variant of Tc4 containing the Tc4 transposase mRNA sequence (*Li and Shaw, 1993*), were not reduced in *simr-1* mutants (*Figure 6A* and *Figure 6—figure supplement 1D*). These data indicate that SIMR-1 is required for small RNA production or maintenance at piRNA-targeted transposons but not at transposons targeted independently of piRNAs.

## *simr-1* mutants have increased levels of small RNAs mapping to histone genes

We next focused on the genes for which the mapped small RNAs increase in *simr-1*, *prg-1* and *mut-16* mutants. In general, fewer genes have a two-fold increase in small RNAs compared to a two-fold decrease in small RNAs for *simr-1*, *prg-1* and *mut-16* mutants at 20°C (*Figure 5A*). These data would indicate that the SIMR-1, along with PRG-1 and MUT-16, plays a more significant role in production or maintenance of small RNAs rather than in suppression of small RNA production. Nonetheless, 213 genes gain small RNAs by greater than two-fold in *simr-1* mutants, 49% of which also gain small RNAs in *prg-1* mutants (*Figures 5A* and *6B*). Interestingly only three of these genes (1%) also gain small RNAs in *mut-16* mutants (*Figure 6B*). While manually examining the list of genes enriched for small RNAs in both *simr-1* and *prg-1* mutants at 20°C, we noticed that this list included numerous histone genes. Of the 104 genes enriched for small RNAs in both *simr-1* and *prg-1* mutants, 28 are histone genes (*Figure 6C*; *Pettitt et al., 2002*). These 28 genes make up 38% of all *C. elegans* histone genes (*Figure 6C*). An additional 30 histone genes (41%) are enriched for small RNAs in only *prg-1* mutants, and only one histone gene is enriched for small RNAs in both *mut-16* and *prg-1* mutants (*Figure 6C*). Overall, histone genes are highly enriched for small RNAs in both *simr-1* and *prg-1* mutants, though this enrichment is lessened across multiple generations at 25°C, suggesting

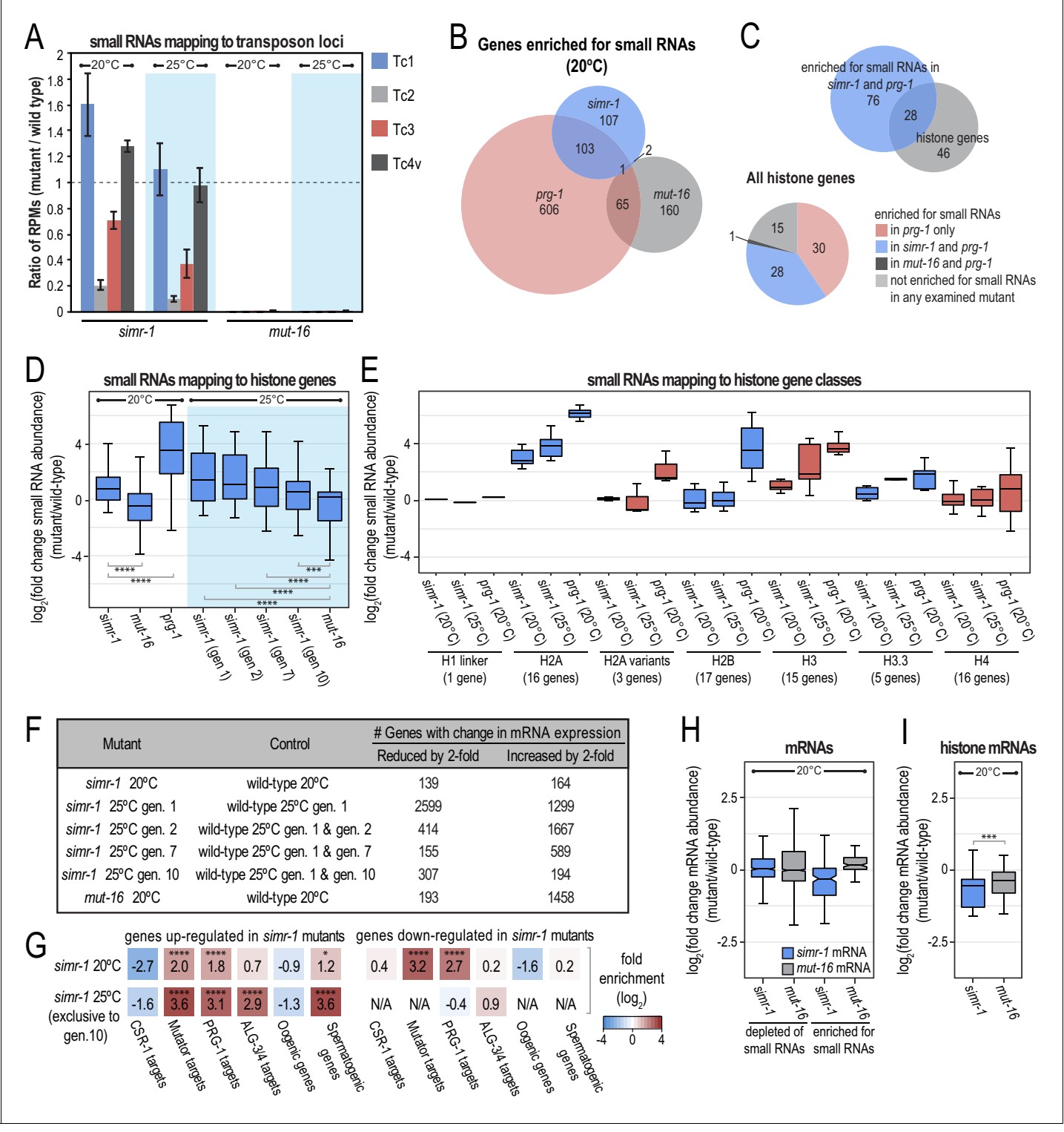

**Figure 6.** *simr-1* mutants display reduced small RNAs mapping to piRNA-dependent transposons and increased small RNAs mapping to histone genes. (A) Ratio of transposon-mapping small RNA reads per million total reads in *simr-1* mutants and *mut-16* mutants raised at 20°C or a single generation at 25°C compared to wild-type shows that small RNAs mapping to Tc2 and Tc3 depend on SIMR-1, but those mapping to Tc1 and Tc4v do not. Error bars indicate standard deviation of two replicate libraries. (B) Venn diagram indicating overlap between genes enriched for small RNAs in *simr-1* mutants, *prg-1* mutants, and *mut-16* mutants. (C) Venn diagram (top) of the 104 genes enriched for small RNAs in both *simr-1* and *prg-1* mutants compared to a list of all histone genes. A pie chart (bottom) of all histone genes shows the number of genes enriched for small RNAs in *prg-1, simr-1,* and *mut-16* mutants compared to wild-type. (D) Box plot displays small RNAs mapping to histone genes in *simr-1* mutants raised at 20°C, a single generation at 25°

*Figure 6 continued on next page*

Figure 6 continued

C, or two, seven, or 10 generations at 25°C, *mut-16* mutants at 20°C or 25°C, and *prg-1* mutants at 20°C compared to wild-type animals at the same temperature and generation, demonstrating that small RNAs mapping to histone genes increase at all temperatures in *simr-1* mutants and in *prg-1* mutants but not *mut-16* mutants. (E) Box plot displays small RNAs mapping to histone gene classes in *simr-1* mutants raised at 20°C or a single generation at 25°C and *prg-1* mutants at 20°C compared to wild-type animals, demonstrating that small RNAs mapping to some histone gene classes increase in both *simr-1* mutants and in *prg-1* mutants while others increase only in *prg-1* mutants. (F) Table indicating the number of genes for which the mRNA expression is either increased or reduced by at least two-fold for each indicated mutant. All genes met the requirements of having a DESeq2 adjusted p-value of ≤0.05 but no minimum read count was required. (G) Enrichment analysis (log$_2$(fold enrichment)) examining the overlap of genes up and down-regulated in *simr-1* mutants with known targets of the CSR-1, *mutator*, PRG-1 and ALG-3/4 small RNA pathways and oogenesis and spermatogenesis-enriched genes. Color of boxes correlates with fold enrichment (red) or depletion (blue). Statistical significance for enrichment was calculated using the Fisher's Exact Test function in R. (H) Box plot displays mRNA expression in *simr-1* (blue) or *mut-16* (grey) relative to wild-type for genes that are enriched or depleted of small RNAs in the same mutants. (I) Box plot displays histone mRNA expression in *simr-1* (blue) or *mut-16* (grey) relative to wild-type, demonstrating that histone mRNA expression is reduced in *simr-1* mutant animals. n.s. denotes not significant and indicates a p-value>0.05, * indicates a p-value≤0.05, ** indicates a p-value≤0.01, *** indicates a p-value≤0.001, **** indicates a p-value≤0.0001. See *Supplementary file 8* for more details regarding statistical analysis.

The online version of this article includes the following figure supplement(s) for figure 6:

**Figure supplement 1.** Small RNAs mapping to piRNA target transposons are reduced and small RNAs mapping to histone genes are increased in *simr-1* mutants.

that it may not be associated with the sterility phenotype (*Figure 6D*). Nonetheless, this enrichment of small RNAs at histone genes in both *simr-1* and *prg-1* mutants is clearly in contrast to *mut-16* mutants at 20°C and 25°C, where the majority of histone genes are unchanged or depleted of small RNAs (*Figure 6D*). We further examined the histone genes by histone gene class and we observed that some histone genes classes such as H2A and H3 genes are enriched for small RNAs in both *simr-1* and *prg-1*, whereas others such as H2B are enriched for small RNAs primarily in *prg-1* mutants (*Figure 6E* and *Figure 6—figure supplement 1E*). This increase in small RNA production to histone genes has been observed previously in *prg-1* mutants and these histone-derived small RNAs are dependent on the *mutator* complex for their biogenesis (*Barucci et al., 2020*; *Reed et al., 2020*). These data suggest that enrichment of small RNAs at certain classes of histone genes is a signature unique to the *simr-1* and *prg-1* mutants and not the *mutator* pathway, and thus provides additional evidence that SIMR-1 plays a key role in the piRNA pathway.

## Most SIMR-1-target genes are not desilenced in a *simr-1* mutant

To determine whether the observed changes to small RNA levels alter gene expression in *simr-1* mutants, we next sequenced mRNAs isolated from wild-type, *simr-1* mutant and *mut-16* mutant animals at 20°C and from wild-type and *simr-1* mutant animals after one, two, seven, or 10 generations at 25°C. We identified 139 genes whose mRNA expression was reduced by at least two-fold in *simr-1* mutants at 20°C and 164 genes whose mRNA expression was increased by at least two-fold in *simr-1* mutants at 20°C (*Figure 6F* and *Supplementary file 5*). Not surprisingly, the *simr-1* up-regulated genes were enriched for *mutator*-target genes and PRG-1-target genes, which initially suggested to us that there may be a direct correlation between loss of small RNAs and an increase in mRNA expression at some loci (*Figure 6G*). However, when we directly compared the list of genes with increased mRNA expression in a *simr-1* mutant (164 genes) to the genes with reduced small RNAs in a *simr-1* mutant (817 genes) we found only 18 genes in common and, furthermore, we do not see a significant change in mRNA expression for the genes depleted of small RNAs in *simr-1* mutants (*Figure 6H* and *Figure 6—figure supplement 1F*). Similarly, in *mut-16* mutants, we do not observe a substantial change in mRNA expression for the genes depleted of small RNAs (*Figure 6H*), which is consistent with recent findings that the majority of *mutator*-target genes and PRG-1-target genes are not desilenced in *mut-16* or *prg-1* mutants, respectively (*Barucci et al., 2020*; *Reed et al., 2020*). We also observed a modest enrichment of spermatogenic genes among the *simr-1* up-regulated genes. This result is similar to the previously published observation that spermatogenesis genes are upregulated in *prg-1* and *mut-16* mutants (*Reed et al., 2020*; *Rogers and Phillips, 2020*), and is consistent with *simr-1* acting with *prg-1* in the regulation of PRG-1 target genes. These data indicate that the majority of SIMR-1-target genes are not derepressed in a *simr-1* mutant, which suggests that either SIMR-1-dependent siRNAs are required only to initiate

but not maintain silencing of their targets or that additional layers of regulation maintain silencing of these genes in the absence of SIMR-1-dependent siRNAs.

We next focused on the genes down-regulated in *simr-1* mutants and found that those genes were also enriched for *mutator*-target genes and PRG-1-target genes (*Figure 6G*), indicating that some *mutator* and PRG-1-target genes are up-regulated, while others are down-regulated in *simr-1* mutants. When we looked exclusively at the genes enriched for small RNAs in *simr-1* mutants, we observed a modest down-regulation of these genes at the mRNA level (*Figure 6H*), indicating that the small RNA gained in the *simr-1* mutant are sufficient to promote down-regulation of their target mRNAs. The same trend was not observed for genes enriched for small RNAs in *mut-16* mutants (*Figure 6H*). Histone genes, including H2A and H3, were amongst those genes enriched for small RNAs and with reduced mRNA expression in *simr-1* mutants (*Figure 6D–E and I*, and *Figure 6—figure supplement 1G*), similar to what has previously been observed in *prg-1* mutants (*Barucci et al., 2020*; *Reed et al., 2020*). We hypothesize that the small RNAs gained in *simr-1* mutants may depend on the *mutator* pathway, similar to what has been shown for the small RNAs targeting histone genes in the *prg-1* mutant (*Barucci et al., 2020*; *Reed et al., 2020*), and therefore these small RNAs are competent to silence their target mRNAs. In contrast, the *mutator* pathway is non-functional in the *mut-16* mutant, therefore the small RNAs gained in this mutant are likely to be a distinct class of small RNAs, possibly CSR-1-class siRNAs, which do not generally silence their mRNA targets (*Claycomb et al., 2009*; *Wedeles et al., 2013*).

Finally, to determine whether the sterility observed in *simr-1* mutants raised at 25°C for 10 generations could be attributed to gene expression changes, we looked for mRNAs up or down-regulated in *simr-1* mutants raised at 25°C for 10 generations compared to wild-type raised under the same conditions that were not up or down-regulated in *simr-1* mutants raised at 20°C or in *simr-1* mutants raised at 25°C for only a single generation (exclusive to gen. 10). We identified only 34 genes significantly down-regulated exclusively at generation 10 and 112 genes significantly up-regulated exclusively at generation 10 (*Supplementary file 5*). The genes up-regulated exclusively in *simr-1* mutants after 10 generations were enriched for *mutator*-target genes, PRG-1-target genes, ALG-3/4-target genes and spermatogenic genes while the down-regulated genes were not enriched for any gene list that we examined (*Figure 6G*). While these up-regulated genes are exclusive to 10 generations at 25°C, the classes of enriched genes (*mutator* targets, PRG-1 targets, and spermatogenic genes) are similar to what was observed in *simr-1* mutants at 20°C. While we cannot attribute the sterility observed in these animals directly to the misregulation of any specific genes, we hypothesize that an increase in the expression of spermatogenesis genes during oogenesis, along with the expression of *mutator* and PRG-1-target genes could contribute to germ cell dysfunction.

## SIMR-1 forms foci near *Mutator* foci, P granules and Z granules

P granules, *Mutator* foci, and Z granules are all phase-separated biomolecular condensates which lie adjacent to one another at the nuclear periphery (*Uebel et al., 2018*; *Wan et al., 2018*; *Brangwynne et al., 2009*). From live imaging of fluorescently-tagged SIMR-1 and MUT-16, we observed that SIMR-1 forms foci closely associated with *Mutator* foci (*Figure 1C*), however from this preliminary analysis we were unable to conclude whether they fully colocalized. To first address the spatial relationship between SIMR-1 and MUT-16, we immunostained fluorescently-tagged SIMR-1 and MUT-16. We observed that SIMR-1 foci are closely associated with *Mutator* foci (96.4% of the time with no empty space between fluorescent signals, n = 56 SIMR-1 foci), however they do not fully colocalize suggesting that they are distinct structures (*Figure 7A*). This result is supported by our previous observation that SIMR-1 foci are not disrupted in a *mut-16* mutant, nor are *Mutator* foci disrupted by the *simr-1; hpo-40* double mutant (*Figure 1D*). Furthermore, we have not been able to unambiguously co-immunoprecipitate MUT-16 with SIMR-1, which indicates that, despite our initial identification of SIMR-1 in the MUT-16 IP-mass spectrometry experiment, the physical interaction between these two proteins may be weak or transient.

Both P granules and Z granules are closely associated with *Mutator* foci (*Wan et al., 2018*; *Phillips et al., 2012*), so we next asked whether SIMR-1 foci colocalize with either PGL-1, marking P granules, or ZNFX-1, marking Z granules. SIMR-1 foci are closely associated with both P granules and with Z granules (100% of the time with P granules, n = 56 SIMR-1 foci, and 100% of the time with Z granules, n = 62 SIMR-1 foci). However, we found that SIMR-1 foci do not fully colocalize with either structure, and in some cases multiple SIMR-1 foci can associate with a single focus of another

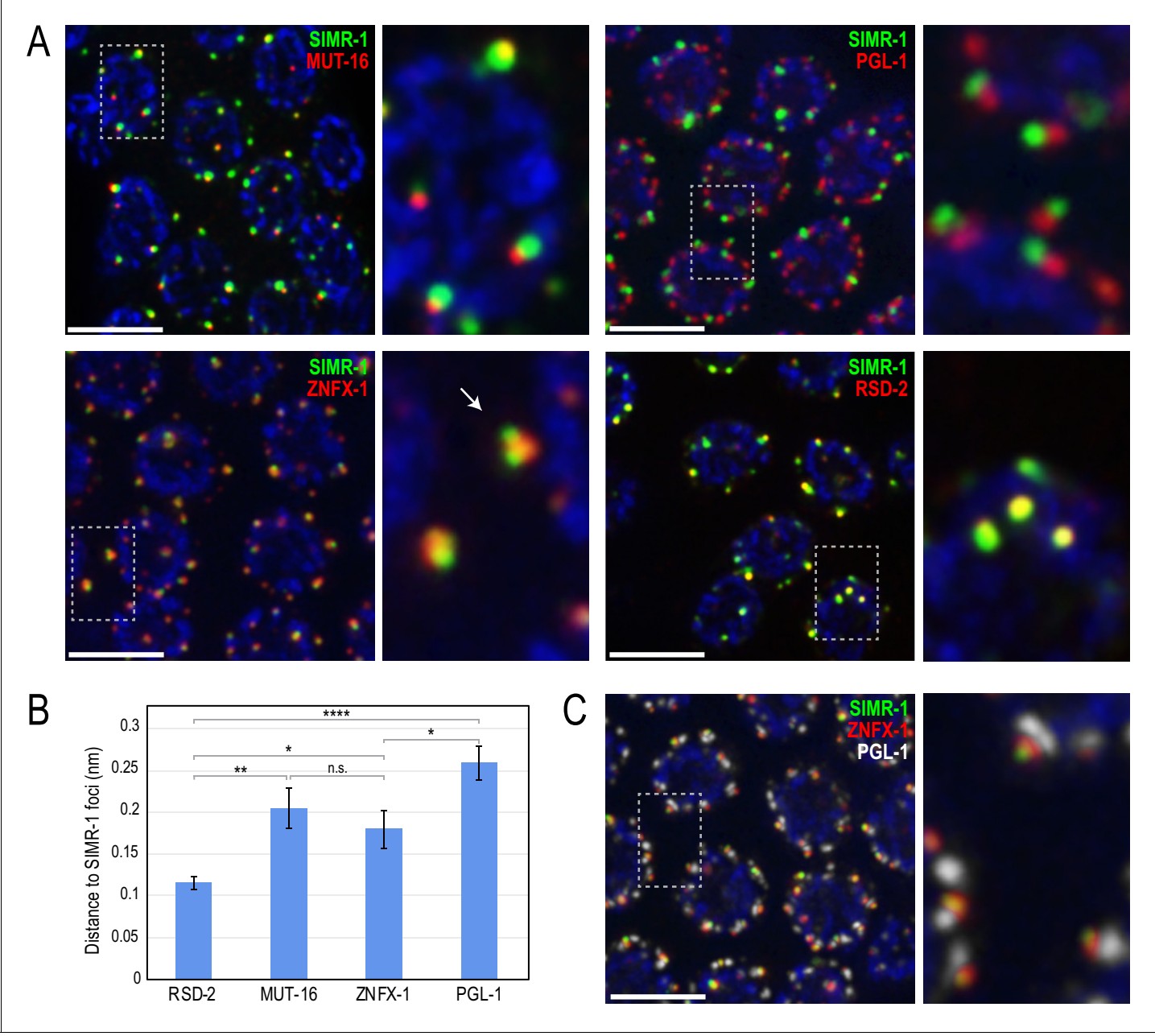

**Figure 7.** SIMR-1 localizes to foci adjacent to P granules and *Mutator* foci. (**A**) Immunostaining of SIMR-1 (green) with MUT-16 (red, top left), PGL-1 (red, top right), ZNFX-1 (red, bottom left), and RSD-2 (red, bottom right) demonstrates that SIMR-1 localizes to foci near *Mutator* foci (MUT-16), P granules (PGL-1), and Z granules (ZNFX-1) but overlaps most substantially with RSD-2 foci. Arrow indicates an example of a single Z granule associated with two SIMR-1 foci. (**B**) Bar graph showing distance between the centers of fluorescence for indicated proteins to SIMR-1 (mean +/- SEM). See Materials and Methods for description of quantification methods. n.s. denotes not significant and indicates a p-value>0.05, * indicates a p-value≤0.05, ** indicates a p-value≤0.01, **** indicates a p-value≤0.0001. See *Supplementary file 8* for more details regarding statistical analysis. (**C**) Immunostaining of SIMR-1 (green), ZNFX-1 (red), and PGL-1 (white) allows for visualization of the stacked SIMR/Z granule/P granule foci. All images are projections of 3D images following deconvolution. DAPI is blue in all panels and scale bars are 5 μm.

The online version of this article includes the following source data and figure supplement(s) for figure 7:

**Source data 1.** Data used to generate *Figure 7B*.

**Figure supplement 1.** SIMR-1 and PRG-1 localize independently and to distinct granules.

granule type (*Figure 7A*, see inset for SIMR-1 and ZNFX-1 localization). SIMR-1 foci do appear to be more closely associated with Z granules than with P granules and quantification of distances between fluorescence centers of each foci supports this observation (*Figure 7A–B*). Because SIMR-1 promotes siRNA biogenesis at piRNA target genes, we also examined the colocalization of SIMR-1 and PRG-1, which has previously been shown to localize to P granules (*Batista et al., 2008*; *Wang and Reinke, 2008*). Similar to what we observed with PGL-1, PRG-1 is localized adjacent to but not coincident with SIMR-1 foci (*Figure 7—figure supplement 1A*). Furthermore, SIMR-1 is not required for PRG-1 localization or expression, and PRG-1 is not required for SIMR-1 localization (*Figure 7—figure supplement 1B–C*). These data indicate that, while SIMR-1 and PRG-1 function in the same pathway to mediate siRNA biogenesis at piRNA target genes, they do not colocalize and are not required for one another's localization or expression.

Also identified in our MUT-16 and SIMR-1 immunoprecipitations was RSD-2, a previously characterized RNAi factor required for exogenous RNAi introduced in low doses and production of secondary siRNAs at target genes dependent on the ERGO-1 primary siRNA pathway. Because RSD-2 also forms foci in close proximity to *Mutator* foci (*Figure 1—figure supplement 1C*), we next generated a strain with fluorescently-tagged SIMR-1 and RSD-2. Following immunostaining, we observed that SIMR-1 and RSD-2 were highly coincident, suggesting SIMR-1 and RSD-2 may localize to the same perinuclear structure (*Figure 7A–B*). These results indicate that SIMR-1 and RSD-2 interact closely with one another at perinuclear foci near but distinct from *Mutator* foci, P granules and Z granules. Because these foci are distinct from previously characterized structures, we are calling them SIMR foci.

Finally, to better understand the organization of these multiple perinuclear foci, we immunostained for SIMR-1, ZNFX-1, and PGL-1 together. Interestingly, we observed that the foci appeared to be stacked, with ZNFX-1 localizing between SIMR-1 and PGL-1 (*Figure 7C*). This result is reminiscent of the tripartite PZM granule (P granule/Z granule/*Mutator* foci) observed by *Wan et al. (2018)*, except that we observe the Z granule flanked by SIMR foci and P granules, instead of *Mutator* foci and P granules. Therefore, our data suggest that there are at least four separate compartments at the nuclear periphery in *C. elegans* germ cells, that together constitute *C. elegans* nuage, each with unique protein components and a distinct molecular role in the RNA silencing pathway.

## Discussion

*C. elegans* utilize the highly abundant siRNAs synthesized by the *mutator* complex to reinforce silencing initiated by the piRNA pathway. Here we identify a Tudor domain protein, SIMR-1, required to mediate effective production of siRNAs from many piRNA-target mRNAs. We demonstrate that SIMR-1 has a phenotype similar to that of PRG-1, in that *simr-1* mutants can desilence a sensitized piRNA sensor and SIMR-1 is required to prevent sterility after reestablishing WAGO-class 22G-siRNA production. However, the phenotypes associated with *simr-1* are often weaker than those of *prg-1* (see *Figures 4B*, *5C–D* and *6B–E*), suggesting that *simr-1* is not absolutely required to mediate siRNA amplification at all piRNA target genes, or it acts cooperatively with other pathways or proteins. SIMR-1 is not RNAi-defective, it cannot desilence a piRNA sensor silenced by RNAe, and it cannot desilence the ERGO-1-dependent siRNA sensor, all phenotypes associated with the downstream *mutator* pathway. Furthermore, siRNAs are reduced at many piRNA-target loci in *simr-1* mutants, but piRNAs themselves are unaffected. Like PRG-1 and the *mutator* complex, SIMR-1 forms foci near the nuclear periphery of germ cells, and while these perinuclear condensates are adjacent to one another, they all appear to be distinct substructures. Thus, our work identifies a novel player acting at a step in between piRNA biogenesis and siRNA amplification by *mutator* complex and suggests a role for multiple perinuclear condensates to promote piRNA-mediated siRNA production.

### Tudor domain proteins in piRNA-mediated silencing

Tudor domain proteins are thought to act as scaffolds in the piRNA pathway, to engage and assemble multiple partner proteins (*Pek et al., 2012*). Through promotion of protein-protein interactions, they can drive piRNA biogenesis and piRNA target silencing. For example, the *Drosophila* Tudor domain protein, Krimper, interacts with two Piwi proteins, Aubergine and Ago3, to coordinate assembly of the ping-pong processing complex (*Webster et al., 2015*). Of note, sDMA of Aubergine is required for interaction with Krimper, but Ago3 can interact with Krimper independently of

sDMA, emphasizing that Tudor domain proteins can play critical roles in the piRNA pathway independent of sDMA. In fact, like SIMR-1, many human and *Drosophila* Tudor domain proteins carry mutations in aromatic cage residues, indicating they may have lost the ability to bind methylated arginine substrates (*Handler et al., 2011*; *Zhang et al., 2017*). For example, mammalian Tudor domain protein, TDRD2, which is missing one of the four aromatic cage residues, preferentially recognizes an unmethylated peptide of PIWIL1 over a dimethylated peptide. This recognition occurs through a negatively charged groove that occurs at the interface of the canonical Tudor domain and the flanking conserved elements making up the extended Tudor domain (*Zhang et al., 2017*). This data would suggest that Tudor domain proteins that are missing the aromatic cage residues, like SIMR-1, may still make functional interactions with Piwi proteins or other small RNA pathway proteins. They may mediate interactions preferentially with unmethylated substrates and/or compete with other Tudor domain proteins for substrates dependent on methylation status. While we have not yet identified the relevant protein-binding partners of the SIMR-1 Tudor domain, we hypothesize that they are likely members of the piRNA pathway or *mutator* complex and contain the RG repeat motif preferentially recognized by the Tudor domain. An obvious candidate is PRG-1 itself, as it contains a 'GRGRGRG' sequence near its N-terminus, however there are certainly other candidates and further experiments will be necessary to test this possibility.

## Regulation of piRNA-target genes by perinuclear condensates

While we do not have direct evidence for a physical interaction between SIMR-1 and PRG-1, it is likely that SIMR-1 interacts with either PRG-1 or some other member of the piRNA pathway to promote the downstream regulation of piRNA target mRNAs by the *mutator* complex. Similarly, because SIMR-1 was initially identified in a MUT-16 immunoprecipitation, it may also interact directly with the *mutator* complex, even if transiently. It is therefore interesting that PRG-1, SIMR-1, and the *mutator* complex all localize to distinct sub-compartments of nuage (*Supplementary file 1*). We have observed that Z granules localize between SIMR-1 foci and P granules, similar to the organization of *Mutator* foci, Z granules and P granules (*Wan et al., 2018*). We have not been able to image SIMR foci with *Mutator* foci, Z granules, and P granules simultaneously, so it remains to be determined how these four substructures assemble together and whether SIMR foci bridge *Mutator* foci and Z granules, *Mutator* foci bridge SIMR foci and Z granules, or whether all three interact. *Mutator* foci, P granules, and Z granules all assemble through intracellular phase separation, which brings about the question as to whether SIMR foci may also behave in a liquid-like manner. While we have not tested this formally, the localization of SIMR-1 nestled among these three other biomolecular condensates is certainly suggestive. The dynamic nature of these various condensates could facilitate exchange of RNAs or protein components between compartments, which may explain how piRNA pathway proteins, SIMR-1, and the *mutator* complex could occupy distinct substructures while facilitating regulation of the same mRNA target genes. Perhaps some proteins have properties making them immiscible in multiple condensates allowing them to promote transfer of RNAs between compartments, or alternatively, the exchange of RNAs may occur at their interface.

## RSD-2 and SIMR-1 promote the interaction between distinct primary and secondary siRNA pathways

The colocalization of SIMR-1 and RSD-2 is somewhat surprising given that SIMR-1 and RSD-2 act in distinct small RNA pathways. Specifically, SIMR-1 acts downstream of PRG-1 in the piRNA pathway and has no defects in exogenous RNAi, whereas RSD-2 is required to mount an efficient response to exogenous RNAi and silence ERGO-1-target genes, but is not required for the production of secondary siRNAs at piRNA target genes (*Han et al., 2008*; *Zhang et al., 2012*). While many of their targets are distinct, SIMR-1 and RSD-2 may play similar roles in mediating the interaction between primary and secondary siRNA pathways and thus their colocalization may be indicative of a subcellular compartment mediating this transition between primary and secondary small RNA pathways.

Like RSD-2, the Tudor domain protein RSD-6, the Maelstrom domain protein RDE-10, the RING-type zinc finger protein RDE-11, and the DEAD box ATPase and Vasa ortholog RDE-12 likely act downstream of primary Argonaute proteins RDE-1 and ERGO-1 and are required for the accumulation of *mutator*-dependent secondary siRNAs (*Zhang et al., 2012*; *Shirayama et al., 2014*; *Yang et al., 2014*; *Yang et al., 2012*). Interestingly, there is no data to suggest that any of these

proteins act with SIMR-1 downstream of PRG-1, suggesting that there could be a completely different set of factors that interact with SIMR-1 at piRNA targets. While no localization has been determined for RDE-10 and RDE-11, RSD-6 localizes to foci near P granules that may be coincident with SIMR-1 foci. RDE-12 localizes to both RSD-6 foci and P granules, suggesting it can traverse the boundary between perinuclear condensates, and it has been proposed that RDE-12 may shuttle primary siRNA bound target mRNAs from P granules to RSD-6 foci to initiate *mutator*-dependent siRNA synthesis (*Yang et al., 2014*). While loss of RDE-12 does not affect siRNAs mapping at piRNA target genes, there are 36 RDE-12 paralogs in *C. elegans,* several of which localize at or near P granules, including GLH-1,–2, −3,–4, DDX-19, LAF-1, MUT-14, and VBH-1 (*Supplementary file 1*). One of these proteins could potentially serve a function similar to RDE-12, in the shuttling of piRNA-targeted mRNAs into SIMR-1 foci and ultimately to *Mutator* foci.

In conclusion, numerous proteins have been identified in *C. elegans* that are required for piRNA transcription, trimming, and modification (*Kamminga et al., 2012*; *Montgomery et al., 2012*; *Billi et al., 2012*; *Tang et al., 2016*; *Weick et al., 2014*; *de Albuquerque et al., 2014*; *Kasper et al., 2014*; *Cordeiro Rodrigues et al., 2019*; *Zeng et al., 2019*), however, how mRNAs travel between the piRNA pathway, required for mRNA recognition, to the *mutator* pathway, necessary for siRNA production has remained a mystery. Here we demonstrate that the Tudor domain protein, SIMR-1, is required at a step between the piRNA pathway to the *mutator* complex. SIMR-1 may function similarly to how Krimper coordinates Ago3 and Aubergine during ping pong piRNA biogenesis in *Drosophila* (*Webster et al., 2015*), but in this case bridging the gap between the primary and secondary phases of the *C. elegans* piRNA silencing pathway. Finally, SIMR-1 localizes to cytoplasmic foci near P granules, Z granules, and *Mutator* foci, implicating a series of distinct perinuclear condensates in the regulation of mRNAs by the piRNA pathway and *mutator* complex.

## Materials and methods

### Strains

The *C. elegans* wild-type strain used is N2. Worms were raised at 20˚C according to standard conditions unless otherwise stated (*Brenner, 1974*). Mutants generated by CRISPR or obtained from the CGC were outcrossed prior to sequencing or other analysis. All strains used for this project are listed in *Supplementary file 6* (key resources table).

### Plasmid and strain construction

All GFP, mKate2, or mCherry tagged strains were generated by CRISPR genome editing, with tags inserted at the endogenous locus. *simr-1::gfp::3xFLAG* repair template was assembled into pDD282 and *mKate2::3xMyc::prg-1* repair template was assembled into pDD287 (Addgene plasmid # 66823 and #70685) according to published protocols (*Dickinson et al., 2015*). Design of the *mCherry::2xHA* plasmid was described previously (*Uebel et al., 2018*). The *mCherry::2xHA* region, which include intronic Floxed *Cbr-unc-119(+)*, was amplified by PCR and assembled by isothermal assembly with ~1 kb of sequence from either side of the stop codon of the gene to be tagged and the XhoI/EagI digested pBluescript vector (*Gibson et al., 2009*). A similar method was used to generate CRISPR-mediated deletions. A region containing the Floxed *Cbr-unc-119(+)* was amplified from the *mCherry::2xHA* plasmid and assembled by isothermal assembly with ~500 bp - 1 kb of sequence from near the start and stop codon of the gene to be deleted and the XhoI/EagI digested pBluescript vector. Primers used to amplify homology arms are listed in *Supplementary file 7*. To protect the repair template from cleavage, we introduced silent mutations at the site of guide RNA targeting by incorporating these mutations into one of the homology arm primers or, if necessary, by performing site-directed mutagenesis (*Dickinson et al., 2013*). All guide RNA plasmids were generated by ligating oligos containing the guide RNA sequence into BsaI-digested pRB1017 (Addgene plasmid # 59936) (*Arribere et al., 2014*). Guide RNA sequences are provided in *Supplementary file 7*. For the introduction of the R159C mutation in *SIMR-1::gfp::3xFLAG*, we used an oligo repair template and RNA guide (*Supplementary file 7*).

CRISPR injections were performed according to published protocols (*Dickinson et al., 2013*; *Dickinson et al., 2015*; *Ward, 2015*; *Arribere et al., 2014*; *Paix et al., 2015*; *Dokshin et al., 2018*). CRISPR injection mixes included 10–25 ng/µl repair template, 50 ng/µl guide RNA plasmid, 50 ng/µl

*eft-3p::cas9-SV40_NLS::tbb-2 3'UTR* (Addgene plasmid # 46168) or *eft-3p::cas9::tbb-2 3'UTR* (Addgene plasmid # 61251), 2.5–10 ng/µl GFP or mCherry co-injection markers, and 10 ng/µl *hsp-16.1:: peel-1* negative selection (pMA122, Addgene plasmid # 34873). mCherry constructs were injected into USC715: *mut-16(cmp3[mut-16::gfp::3xFLAG + loxP]) I; unc-119(ed3) III*. Deletion constructs were injected into HT1593: *unc-119(ed3)*, except for the *hpo-40* deletion construct, which was injected directly into *simr-1(cmp36) I; unc-119(ed3) III*. For some strains, floxed *Cbr-unc-119(+)* cassettes were excised using *eft-3p::Cre* (pDD104, Addgene plasmid # 47551) (*Dickinson et al., 2013*), however we observed no discernable increase in mCherry-tagged protein expression after *Cbr-unc-119(+)* cassette excision. *SIMR-1::gfp::3xFLAG* and *mKate2::3xMyc::prg-1* was injected into the wild-type strain. For the R159C mutation of SIMR-1, the injection mix included 0.25 µg/µl Cas9 protein (IDT), 100 ng/µl tracrRNA, 14 ng/µl *dpy-10(cn64)* crRNA, 42 ng/µl *simr-1* crRNA, and 110 ng/µl of each repair template, and was injected into USC1022(*simr-1(cmp112[simr-1::GFP + loxP + 3xFLAG]) I*) (*Paix et al., 2015*; *Dokshin et al., 2018*).

## Mass spectrometry

~500,000 synchronized N2 (wild-type) or USC717 (*mut-16(cmp3[mut-16::gfp::3xFLAG + loxP])*) adult *C. elegans* (~68 hr at 20°C after L1 arrest) were collected in IP Buffer (50 mM Tris-Cl pH 7.4, 100 mM KCl, 2.5 mM MgCl$_2$, 0.1% Igapal CA-630, 0.5 mM PMSF (0.5 mM), cOmplete Protease Inhibitor Cocktail (Roche 04693159001), and RNaseOUT Ribonuclease Inhibitor (ThermoFisher 10777019)), frozen in liquid nitrogen, and homogenized using a mortar and pestle. After further dilution into IP buffer (1:10 packed worms:buffer), insoluble particulate was removed by centrifugation and a sample was taken as 'input.' The remaining lysate was used for the immunoprecipitation. GFP and FLAG immunoprecipitation was performed at 4°C for 2 hr using anti-GFP affinity matrix [RQ2 clone] (MBL International D153-8) and anti-FLAG affinity matrix [M2 clone] (Sigma-Aldrich A2220), then washed 10 times in immunoprecipitation buffer. After immunoprecipitation, samples were precipitated using the ProteoExtract Protein Precipitation Kit (EMD Millipore 539180) and submitted to the Taplin Mass Spectrometry facility at Harvard Medical School for protein identification.

## Antibody staining and imaging

Live imaging was conducted by dissecting *C. elegans* animals in M9 buffer containing sodium azide and imaging immediately following dissection. For immunofluorescence, worms were dissected in egg buffer containing 0.1% Tween-20 and fixed in 1% formaldehyde in egg buffer as described (*Phillips et al., 2009*). Samples were immunostained with mouse anti-FLAG (M2, Sigma F1804), rat anti-HA (3F10, Sigma 11867423001), and mouse anti-PGL-1 (DSHB AB 531836). Alexa-Fluor secondary antibodies were purchased from ThermoFisher Scientific. All worms were dissected as one-day-old adults (~24 hr after L4). Imaging was performed on a DeltaVision Elite microscope (GE Healthcare) using 60x N.A. 1.42 oil-immersion objective. When data stacks were collected, deconvolution was performed using the SoftWoRx package and presented as maximum intensity projections. Images were pseudocolored using Adobe Photoshop.

For scoring of apoptotic germ cells, corpses were identified using the *bcIs39* (CED-1::GFP) reporter, which is expressed in gonadal sheath cells and can be observed surrounding germ cell corpses during engulfment. A minimum of 20 gonads arms were scored per genotype and condition. Information regarding statistical analysis provided in *Supplementary file 8*.

Quantification of distance between foci centers was performed in ImageJ according to published methods (*Wan et al., 2018*). We imaged pachytene germ cell nuclei from two animals. Three granules selected from each of four germ cells for a total of 12 granules per animal. Z stacks were opened using the 3D object counter plugin for ImageJ to collect the x, y, and z coordinates for the center of each desired foci (*Bolte and Cordelières, 2006*). With these coordinates, distances between the foci centers were calculated using the distance formula, $\sqrt{(x_2 - x_1)^2 + (y_2 - y_1)^2 + (z_2 - z_1)^2}$. To account for chromatic shift between channels, distances were calculated between each pair of channels using TransFluorospheres streptavidin-labeled microspheres, 0.04 µm (ThermoFisher, T10711) and these distances were used to correct granule distances.

### Protein domain identification

The protein alignment of SIMR-1 with HPO-40 (*C. elegans*), CJA21107 (*C. japonica*), CBN15556 (*C. brenneri*), and, CRE08315 (*C. remanei*) was generated using Clustal Omega and cladogram was made in Evolview V3 (*Sievers et al., 2011*; *Subramanian et al., 2019*). The SIMR-1 protein sequence was input into the HHPred server to identify remote protein homologs with structural similarity (*Söding et al., 2005*). A region spanning amino acids 89–264 of the SIMR-1 protein aligned with extended Tudor domain region of multiple Tudor domain proteins. A Clustal Omega protein alignment of this putative extended Tudor domain region of SIMR-1, HPO-40, and their related nematode orthologs was then generated, and this alignment was entered into the HHpred server to improve sensitivity. The top non-redundant identified proteins, their Protein Data Bank ID code, and HHpred E-value were *H. sapiens* TDRD1 (5M9N) – 5.5e-8, *M. musculus* TDRD1 (4B9X) – 4.3e-7, *D. melanogaster* Papi/Tdrd2 (5YGB) – 4.5e-7, *H. sapiens* SND1/TDRD11 (5M9O) – 7.2e-6, *D. melanogaster* Tudor (3NTK) – 2.3e-5, *H. sapiens* TDRKH/TDRD2 (6B57) – 1.6e-5, *D. melanogaster* Tudor-SN (2WAC) – 3.2e-5, and *B. mori* PAPI (5VQH) – 2.0e-4.

### RNAi assays

For RNAi assays, synchronized L1 worms raised at 20°C were fed *E. coli* expressing dsRNA against *pos-1*, *lin-29*, *nhr-23*, *lir-1*, *hmr-1*, and *dpy-13* (*Kamath et al., 2003*). For *pos-1* and *hmr-1*, F1 embryos were scored for hatching three to five days after P0 animals were placed on RNAi bacteria. For *lin-29*, *nhr-23*, *lir-1*, and *dpy-13* animals were scored three days after commencement of feeding RNAi for vulval bursting, larval arrest, larval arrest, and shorter length (Dumpy), respectively.

### Transgenerational fertility and brood size assays

Wild-type and mutant *C. elegans* strains were maintained at 20°C prior to temperature-shift experiments. Animals were shifted to 25°C, or back to 20°C, as L1 larvae. For the brood-size assays, 10 L4 animals were picked to individual plates. A single progeny from each plate was selected and moved to a new plate at L4 stage for the following generation. If one or more of the animals was sterile, progeny were selected from one of the replicate plates to maintain the total number of broods scored for each generation at 10. To score the complete brood, each animal was moved to a fresh plate every day until egg-laying was complete. After allowing the progeny 2–3 days to develop, the total number of animals on each plate was counted.

For assessment of sperm viability, wild-type and *simr-1* mutant males were raised either at 20°C, a single generation at 25°C, or following 10 generations of growth at 25°C, and then mated to *fog-2* females raised at 20°C. Brood sizes were scored for 10 *fog-2* females, each mated to four wild-type or *simr-1* mutant males. Males were generated by heat shock and then maintained as a mating plate at 20°C for multiple generations prior to beginning temperature-shift experiments. Information regarding statistical analysis provided in *Supplementary file 8*.

For assessment of oocyte viability, wild-type and *simr-1* mutant hermaphrodites were raised either at 20°C, a single generation at 25°C, or following 10 generations of growth at 25°C, and then mated to four *pgl-1::gfp* males raised at 20°C. Brood sizes were scored for each of 10 wild-type or *simr-1* mutant hermaphrodites, mated to four *pgl-1::gfp* males. Only plates where all progeny were GFP positive were scored to ensure that the mating had occurred. Information regarding statistical analysis provided in *Supplementary file 8*.

### Reestablishing WAGO-class 22G-siRNA production

The *mutator* pathway was restored to WAGO-class 22G-siRNA-defective animals according to the crossing scheme in *Figure 4B* and as previously described (*Phillips et al., 2015*). The *unc-119* mutation was always present in the parental hermaphrodite strain to allow for unambiguous identification of cross vs. self progeny. F1 animals were singled to individual plates as L4 larvae and scored 2–3 days later for presence or absence of progeny.

### Western blots

For Western blots, proteins were resolved on 4–12% Bis-Tris polyacrylamide gels (ThermoFisher), transferred to nitrocellulose membranes, and probed with anti-FLAG 1:1,000 [M2 clone] (Sigma-

Aldrich F1804), anti-actin 1:10,000 (Abcam ab3280), or anti-Myc 1:1,000 [9E10 clone] (ThermoFisher 13–2500). Secondary HRP antibodies were purchased from ThermoFisher.

## Small RNA and mRNA library preparation

Small RNAs (18 to 30-nt) were size selected on denaturing 15% polyacrylamide gels (BioRad 3450091) from total RNA samples. Small RNAs were treated with 5' RNA polyphosphatase (Epicentre RP8092H) and ligated to 3' pre-adenylated adapter with Truncated T4 RNA ligase (NEB M0373L). Small RNAs were then hybridized to the reverse transcription primer, ligated to the 5' adapter with T4 RNA ligase (NEB M0204L), and reverse transcribed with Superscript III (ThermoFisher 18080–051). Small RNA libraries were amplified using Q5 High-Fidelity DNA polymerase (NEB M0491L) and size selected on a 10% polyacrylamide gel (BioRad 3450051).

For mRNA-seq library preparation, nuclease-free $H_2O$ was added to 7.5 µg of each RNA sample, extracted from whole animals, to a final volume of 100 µL. Samples were incubated at 65°C for 2 min then incubated on ice. The Dynabeads mRNA Purification Kit (ThermoFisher 61006) was used according to the manufacturer's protocol. 20 µL of Dynabeads was used for each sample. 100 ng of each mRNA sample was used to prepare libraries with the NEBNext Ultra II Directional RNA Library Prep Kit for Illumina (NEB E7760S) according to the manual, using NEBNext multiplex oligos for Illumina (NEB E7335S).

Library concentration was determined using the Qubit 1X dsDNA HS Assay kit (ThermoFisher Q33231) and quality was assessed using the Agilent BioAnalyzer. Libraries were sequenced on the Illumina NextSeq500 (SE 75 bp reads) platform.

## Bioinformatic analysis

For small RNA libraries, sequences were parsed from adapters using FASTQ/A Clipper (options: -Q33 -l 17 c -n -a TGGAATTCTCGGGTGCCAAGG) and quality filtered using the FASTQ Quality Filter (options: -Q33 -q 27 p 65) from the FASTX-Toolkit (http://hannonlab.cshl.edu/fastx_toolkit/), mapped to the *C. elegans* genome WS258 using Bowtie2 v. 2.2.2 (default parameters) (*Langmead and Salzberg, 2012*), and reads were assigned to genomic features using FeatureCounts (options: -t exon -g gene_id -O –fraction –largestOverlap) which is part of the Subread v. 1.5.1 package (*Liao et al., 2014*; *Liao et al., 2013*). For all analysis examining total small RNA levels mapping to genes, sequences were assigned to features in a modified version of the WS258 conical gene set GTF file where miRNAs and piRNAs were excluded. For mRNA libraries, sequences were parsed from adapters using Cutadapt v. 1.18 (options: -a AGATCGGAAGAGCACACGTCTGAAC TCCAGTCA -m 17 –nextseq-trim=20 max-n 2) (*Martin, 2011*) and mapped to the *C. elegans* genome WS258 using HISAT2 v. 2.1.0 (options: -k 11) (*Kim et al., 2015*) and the transcriptome using Salmon v. 0.14.1 (options: -l A –validateMappings) (*Patro et al., 2017*). Differential expression analysis was done using DESeq2 v. 1.22.2 (*Love et al., 2014*). For both small RNA and mRNA-seq libraries, a two-fold-change cutoff and a DESeq2 adjusted p-value of ≤0.05 was required to identify genes with significant changes in small RNA or mRNA expression. For small RNA-seq libraries, all genes with differentially-expressed small RNAs also met the requirements of having at least 10 RPM in either wild-type or mutant libraries. *Mutator*-target genes, piRNA-target genes, and ERGO-1-target genes were defined as those whose total mapped small RNA levels were reduced by at least two-fold in *mut-16*, *prg-1*, and *ergo-1* mutants compared to wild-type, respectively, with at least 10 RPM in wild-type samples and a DESeq2 adjusted p-value of ≤0.05. PRG-1-independent *mutator* targets are a subset of the *mutator* targets for which the total mapped small RNA levels in *prg-1* mutants are either unchanged or increased relative to wild-type. CSR-1 target genes, ALG-3/4 target genes, spermatogenesis-enriched genes, and oogenesis-enriched genes were previously described (*Lee et al., 2012*; *Conine et al., 2013*; *Reinke, 2004*). All siRNA target genes are defined as all *C. elegans* genes with at least 10 RPM in wild-type or mutant small RNA libraries. Additional data analysis was done using R, Excel, and custom Python scripts. Venn diagrams were generated using BioVenn (*Hulsen et al., 2008*) and modified in Adobe Illustrator. Reads per million total reads were plotted along the WS258 genome using Integrative Genomics Viewer 2.3.90 (*Thorvaldsdóttir et al., 2013*; *Robinson et al., 2011*). Sequencing data is summarized in *Supplementary file 9*.

## Accession numbers

High-throughput sequencing data for RNA-sequencing libraries generated during this study are available through Gene Expression Omnibus (GSE138220 for preliminary *simr-1* small RNA, and *simr-1* mRNA sequencing data, GSE134573 for *mut-16* small RNA and mRNA sequencing data, and GSE145217 for *prg-1* and *ergo-1* small RNA sequencing data).

## Acknowledgements

We thank the members of the Phillips lab for helpful discussions and feedback on the manuscript. Some strains were provided by the CGC, which is funded by NIH Office of Research Infrastructure Programs (P40 OD010440), and Shohei Mitani of the National BioResource Project of Japan. Next generation sequencing was performed by the USC Molecular Genomics Core, which is supported by award number P30 CA014089 from the National Cancer Institute, and by the USC Genome Core.

## Additional information

### Funding

| Funder | Grant reference number | Author |
|---|---|---|
| National Cancer Institute | K22 CA177897 | Carolyn Marie Phillips |
| National Institute of General Medical Sciences | R35 GM119656 | Carolyn Marie Phillips |
| National Institute of General Medical Sciences | T32 GM118289 | Dieu An H Nguyen |
| National Institute of General Medical Sciences | R35 GM119775 | Taiowa A Montgomery |
| Deutsche Forschungsgemeinschaft | KE1888/1-1 | Rene F Ketting |
| Deutsche Forschungsgemeinschaft | KE1888/1-2 | Rene F Ketting |
| Pew Charitable Trusts | Pew Scholar in the Biomedical Sciences | Carolyn Marie Phillips |
| National Science Foundation | DGE 1418060 | Celja J Uebel |
| University of Southern California | Dornsife College Chemistry-Biology Interface trainee | Celja J Uebel |

The funders had no role in study design, data collection and interpretation, or the decision to submit the work for publication.

### Author contributions

Kevin I Manage, Conceptualization, Formal analysis, Investigation, Writing - review and editing; Alicia K Rogers, Dylan C Wallis, Kristen C Brown, Formal analysis; Celja J Uebel, Funding acquisition, Investigation; Dorian C Anderson, Katerina Arca, Investigation; Dieu An H Nguyen, Resources, Funding acquisition; Ricardo J Cordeiro Rodrigues, Bruno FM de Albuquerque, Resources; René F Ketting, Resources, Funding acquisition, Writing - review and editing; Taiowa A Montgomery, Formal analysis, Funding acquisition, Writing - review and editing; Carolyn Marie Phillips, Conceptualization, Supervision, Funding acquisition, Writing - original draft, Writing - review and editing

### Author ORCIDs

Kevin I Manage (iD) https://orcid.org/0000-0003-1992-0782
Alicia K Rogers (iD) https://orcid.org/0000-0001-5525-6095
Dylan C Wallis (iD) https://orcid.org/0000-0003-2375-5062
Celja J Uebel (iD) http://orcid.org/0000-0002-0362-1238
Dieu An H Nguyen (iD) https://orcid.org/0000-0003-4927-7329

Kristen C Brown (iD) https://orcid.org/0000-0002-5495-3371
Bruno FM de Albuquerque (iD) https://orcid.org/0000-0001-8483-6822
René F Ketting (iD) https://orcid.org/0000-0001-6161-5621
Taiowa A Montgomery (iD) https://orcid.org/0000-0001-7857-3253
Carolyn Marie Phillips (iD) https://orcid.org/0000-0002-6228-6468

**Decision letter and Author response**
Decision letter https://doi.org/10.7554/eLife.56731.sa1
Author response https://doi.org/10.7554/eLife.56731.sa2

## Additional files

### Supplementary files

- Source code 1. Python script to calculate reads per million (RPM) over a genomic region.
- Source code 2. Python script to generate WIG files for visualization of tracks in IGV.
- Supplementary file 1. Known components of the L4 and adult germ granules.
- Supplementary file 2. Proteins identified in MUT-16 IP. Those examined further highlighted in blue.
- Supplementary file 3. Small expression in *simr-1*, *mut-16*, and *prg-1* mutant animals.
- Supplementary file 4. mRNA target predictions for *simr-1*-depleted piRNAs.
- Supplementary file 5. mRNA expression in *simr-1* and *mut-16* mutant animals.
- Supplementary file 6. Key resources table.
- Supplementary file 7. Oligonucleotides sequences used in this study.
- Supplementary file 8. Statistical analysis.
- Supplementary file 9. Sequencing library statistics.
- Transparent reporting form

### Data availability

High-throughput sequencing data for RNA-sequencing libraries generated during this study are available through Gene Expression Omnibus (GSE138220 for preliminary simr-1 small RNA, and simr-1 mRNA sequencing data, GSE134573 for mut-16 small RNA and mRNA sequencing data, and GSE145217 for prg-1 and ergo-1 small RNA sequencing data).

The following datasets were generated:

| Author(s) | Year | Dataset title | Dataset URL | Database and Identifier |
|---|---|---|---|---|
| Manage KI, Rogers AK, Uebel CJ, Anderson DC, Arca K, Brown KC, Montgomery TA, Phillips CM | 2020 | A Tudor domain protein, SIMR-1, promotes siRNA production at piRNA-targeted mRNAs in C. elegans | https://www.ncbi.nlm.nih.gov/geo/query/acc.cgi?acc=GSE138220 | NCBI Gene Expression Omnibus, GSE138220 |
| Rogers AK, Phillips CM | 2020 | RNAi pathways repress reprogramming of C. elegans germ cells during heat stress | https://www.ncbi.nlm.nih.gov/geo/query/acc.cgi?acc=GSE134573 | NCBI Gene Expression Omnibus, GSE134573 |
| Rogers AK, Phillips CM | 2020 | The eri-6/7 gene locus is part of an autoregulatory mechanism that maintains proper levels of 22G-siRNAs in C. elegans | https://www.ncbi.nlm.nih.gov/geo/query/acc.cgi?acc=GSE145217 | NCBI Gene Expression Omnibus, GSE145217 |

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
