## [Decision Letter]

**Acceptance summary:**

The paper provides interesting new insights into small RNA biology and we thank you for sending this story to *eLife*.

**Decision letter after peer review:**

[Editors’ note: the authors submitted for reconsideration following the decision after peer review. What follows is the decision letter after the first round of review.]

Thank you for submitting your work entitled "A Tudor domain protein, SIMR-1, promotes siRNA production at piRNA-targeted mRNAs in *C. elegans*" for consideration by *eLife*. Your article has been reviewed by two peer reviewers, and the evaluation has been overseen by a Reviewing Editor and a Senior Editor. The reviewers have opted to remain anonymous.

Our decision has been reached after consultation between the reviewers. Based on these discussions and the individual reviews below, we regret to inform you that we cannot presently accept the manuscript for publication in *eLife*.

As you will see in the detailed comments by the reviewers below, we appreciate the many interesting insights that your manuscript describes. However, there was an overall concern that, while interesting, the many observations described do not coalesce into a coherent picture at present. Several of the results presented were considered too preliminary and not fully flashed out. You will see that the reviewers propose a number of additional experiments. However, if you are able to successfully address the reviewers’ concerns, *eLife* may be interested in looking at a new version of this manuscript. Since the required revisions are quite substantial, such a manuscript would need to be considered a new submission.

*Reviewer #1:*In *C. elegans*, small RNAs antisense to mRNAs regulate gene expression. The small RNAs are synthesized by RNA-dependent RNA polymerases (RdRPs) that accumulate in perinuclear condensates called mutator foci. How specific mRNAs are chosen for small RNA synthesis is not known, but genomically-encoded piRNAs appear to be involved for at least a subset of loci including transposable elements.

The manuscript by Manage et al., reports on a new component of the *C. elegans* small RNA synthesis machinery. SIMR-1 contains a TUDOR-like domain, a domain often found in Argonaute-interacting proteins. Proteomic analyses of SIMR-1 indicate that SIMR-1 interacts with MUT-16, other Mutator foci proteins, and WAGO-1, an Argonaute that also interacts with MUT-16. Like Mutator foci proteins, SIMR-1 is required for the synthesis and/or maintenance of small RNAs targeting many loci (>600), including ~100 loci also targeted by piRNAs. SIMR-1 localizes to a new granule adjacent to Mutator foci. The authors speculate that SIMR-1 may function in the "handoff" of mRNAs targeted by piRNAs to mutator foci. This conclusion, however, does not seem well supported by the data since SIMR-1 mutants affect many more genes beyond those targeted by piRNAs (point #1). SIMR-1 interacts with the Argonaute WAGO-1, but a functional connection between SIMR-1 and WAGO-1 was not explored (point #5).

1) The authors do a comprehensive analysis of small RNA defects in SIMR-1 mutants raised at different temperatures and identify nearly ~1000 affected loci. Consistent with SIMR-1 interacting with the Mutator scaffold MUT-16, >90% of the genes affected in *simr-1* mutants are also affected in mut-16 mutants. These genes include genes targeted by piRNAs, but the latter represent only a minority (<20%) of genes affected in *simr-1* mutants, so it is not clear why the authors focus so much on those genes.

2) '*simr-1* mutants fail to produce high levels of siRNAs from the majority of piRNA-target loci' Figure 4C seems to contradict this statement since at 20°C only <50% of piRNA-target loci are affected. Also, what criteria were used to define "piRNA-targeted loci"? New analyses by the Mello and Lee labs have identified piRNA binding sites in 100s of mRNAs targeted by PRG-1/piRNA complexes. What happens to those loci in *simr-1* mutants? Are the small RNAs completely gone or reduced? Are small RNAs nearest piRNA-binding sites most or least affected? is there a specific pattern of loss?

3) The authors argue that the defect must be "downstream of the piRNAs" because piRNAs levels overall are not affected in SIMR-1 mutants. They don't report however the levels of the specific piRNAs that target the ~100 piRNA-targeted genes affected in the *simr-1* mutant. The levels of PRG-1 protein (piRNA associated Argonaute) also are not reported.

4) Do the small RNA changes observed in *simr-1* mutants correlate with changes in mRNA levels?

5) SIMR-1 interacts with the Argonaute WAGO-1, but the authors do not explore potential functional connections between SIMR-1 and WAGO-1 – why? Is there a good overlap between WAGO-1-dependent loci and SIMR-1-dependent loci?

6) Figure 6: Systematic quantification of relative granule positions shown is missing. See Wan et al., 2018, for examples of quantification metrics. The authors' findings suggest that they are 4 different types of perinuclear condensates in germ cells – are they all present in a 1:1:1:1 proportion? Is every P granule next to a Z granule next to a SIMR-1 granule next to a mutator focus or does the order and composition vary?

*Reviewer #2:*In this manuscript, Manage et al. have identified a previously uncharacterized protein, called SIMR-1, which interacts with a component of mutator foci, MUT-16, and participates in the accumulation of a subset of endo-siRNAs antisense to piRNA and mutator target genes. They also show that *simr-1* mutant displays a temperature-dependent progressive loss of fertility similarly to mutation in downstream components of the piRNA pathway. Interestingly, they have shown that SIMR-1 has a perinuclear localization in a region distinct from p-granule, z-granule or mutator foci, and that its localization is independent from mutator foci. Moreover, they have identified by sequence alignment an extended Tudor domain protein which might potentially interact with methylated PIWI protein. Based on these results, they propose that SIMR-1 constitutes a distinct germ-granule that acts downstream of PIWI and piRNAs to transport mRNAs targeted by piRNAs from p-granule to mutator foci (the granule involved in the amplification of endo-siRNAs). Overall, the distinct perinuclear localization of this protein is exciting, however there are no experiments that substantiate the claim that this protein has a role in transporting mRNAs form piRNA pathway to downstream mutator foci nor that the putative Tudor domain is functional. I also express some concerns about the methods, data analysis, and interpretation of some key results as listed below.

A) The authors performed a MUT-16 IPs coupled with mass spectrometry to identify novel interacting factors of mutator complex. My first concern is related to the fact that the authors did not specify how many IP replicates they have used and which statistical methods they have performed to identify specific interactors in MUT-16 IPs versus control IPs. It appears that they have considered interacting proteins that have peptides only in MUT-16 GFP and FLAG IPs and not in the control. This approach can limit the number of interacting proteins identifiable in the IPs. Also, the number of peptides they have identified for SIMR-1 is extremely low (2 peptides in FLAG IPs and 1 peptide in GFP IPs) rising doubt that SIMR-1 is a real interactor of MUT-16. Because of the lack of statistical tools used to identify interacting proteins I suggest them to perform Co-IP experiments to confirm this interaction. It is also not known whether this interaction is direct or mediated by RNA, and an experiment such as Co-IP can help to distinguish the nature of their interactions. Also, in Figure 1A they have included the proteins RSD-2, WAGO-1, and MATH-33, which have been identified following another (not clear) approach (their peptides are not uniquely present in MUT-16 IPs). Therefore, it should be removed from this table. Again, if they want to take in consideration such cases they should perform a quantitative proteomic approach, using more replicates and statistical tools.

B) There are no functional data or structural information to asses that the conserved residues contained in SIMR-1 constitute a Tudor domain capable of recognizing a methylated substrate such as PIWI. Moreover, SIMR-1 IP and mass spec experiments failed to identify PRG-1/PIWI protein. The major interactor identified by mass spec is instead RSD-2, which they show to colocalize in the same perinuclear foci as SIMR-1, although it appears to have a different function (RSD-2 is involved in RNAi and not 22G-RNA amplification from piRNA targets and SIMR-1 is not involved in RNAi but in the accumulation of 22G-RNAs from piRNA targets). Therefore, the distinct localization of SIMR-1 per se does not implicate that SIMR-1 mediates the handoff of mRNAs from the piRNA pathway to the downstream Mutator components. Also, the peptide counts of the putative interactors of SIMR-1 shown in Supplementary Figure 6 are extremely low, a factor that rises concerns on the quality of the IP/mass spec experiments. Again, it is not specified how many replicates have been performed. Therefore, some of these interactions, such as MUT-16/SIMR-1 should be confirmed by Co-IP.

C) The authors performed a small RNA-seq analysis to check whether specific population of small RNAs are affected in *simr-1* mutant. They have performed profiles of small RNAs in different conditions (normal temperature of grow or 25°C at several generations), perhaps to correlate gene expression changes with the temperature-dependent progressive loss of fertility phenotype. However, the lack of corresponding RNA-seq profiles does not allow to investigate the effect of temperature on gene expression and how it correlates with the progressive sterility phenotype. Moreover, the most dramatic changes in small RNAs in SIMR-1 mutant occur at 25°C after just one generation. Thus, there is no correlation between progressive loss of fertility and changes in small RNAs. I wonder then if the results across generation of SIMR-1 mutant grown at 25°C are relevant for the current manuscript at all. Therefore, the only conclusion from all the analyses performed is that SIMR-1 is somehow participating in the accumulation of a subset of mut-16 and piRNA targets. The effect of the temperature on small RNA biogenesis is instead not clear.

D) In addition, the small RNA analysis they have performed is not very compelling. For instance, in some cases is not clear if the criteria they have used to define genes with decrease or increase small RNAs. Also, the methods don't clearly show which small RNAs they are considering (22G-RNAs, 26G-RNAs, sense, antisense, broad range of small RNAs 21-26nt, etc.). Because these are small RNAs cloned from total RNAs and not from AGO IPs, the authors cannot be sure that the increases or decreases of small RNAs observed belong to a specific AGO pathway (it is known that mutation in some small RNA biogenesis factor causes misrouting of small RNAs into AGOs). In general, it appears that mutation in SIMR-1 causes a general downregulation (not completely loss) of small RNAs, which are considered to be mut-16 dependent (including some previously defined piRNA targets). Among these affected genes, the piRNA targets appear more affected than the whole mut-16 targets.

E) The authors claim that SIMR-1 acts downstream of the piRNA pathway, however they haven't performed any experiment to prove that piRNA silencing is impaired in *simr-1* mutant. They should use the canonical piRNA sensor strain (Ashe et al., 2012), and check whether the silencing of the piRNA sensor reporter is impaired in *simr-1* mutant. This experiment goes well with the other assays they have used to show that RNAi is not affected in *simr-1* mutant.

Specific comments:

1) Add in Figure 2C the statistical significance of those changes.

2) Add in Figure 2D genes that are statistically significant up-regulated and down-regulated (or >2-fold change). They can label the significant dots with different colours as well. Same in 2E, if any.

3) In Figure 4C the total number of genes belonging to piRNA, ERGO-1, and mut-16 is changing in the different Venn diagram. This is not clear. Shouldn't be constant?

4) Subsection “*simr-1* mutants have increased levels of germ cell apoptosis”, the paragraph title is misleading. The level of germline apoptosis in SIMR-1 is similar to WT at 20°C and lower than WT at 25°C (first generation). It is slightly higher than the corresponding WT at generation 2, 10, 11 and not changing at generation 4,7. Importantly, the WT at 25°C (1 generation) has the highest level of germline apoptosis compared to all the other data points and yet is still fertile. Therefore, the reported level of germline apoptosis is not correlated with the sterility phenotype. This is why I disagree with the Authors' interpretation of the results shown in Figure 3E and the conclusion they wrote in the subsection “*simr-1* mutants have increased levels of germ cell apoptosis”.

5) The first paragraph of the subsection “SIMR-1 is required for small RNA production at piRNA target genes”, contains results that are not shown in any figures. Please, provide a figure that reports these results. Also, it is not written which criteria they have used to identify changes in small RNA populations (fold changes, p-value?), nor it is known from the Materials and methods or text which small RNAs they are considering (22G-RNAs, 26G-RNAs, sense, antisense, broad range of small RNAs 21-26nt, etc.). Also, do the small RNA-depleted genes at 25°C (which are not depleted in subsequent generations) reacquire the small RNAs in progressive generations?

6) The results from the analyses shown in the subsection “*simr-1* mutants have increased levels of siRNAs mapping to histone genes”, are also not clear and maybe not relevant for the current manuscript. The results described lack a figure reference. Are they shown in Supplementary Figure 5B? Also, in Supplementary Figure 5B they only show depleted siRNAs and not enriched siRNAs. Also, they only consider lists of genes depleted of 22G-RNAs. Maybe if they consider log2(fold changes) they will find more insightful results. In regard to the small RNAs targeting histone mRNAs they have cited a submitted paper (not sure if it is possible to cite the journal where the paper is submitted) and conclude this might be relevant. Mutation in MUT-16, which is linked to piRNA pathway, do not show the same histone small RNAs. Moreover, histone small RNAs are present at all temperature, therefore also do not correlate with the progressive loss of fertility in SIMR-1 mutant. Again, I am not sure how this paragraph fits with the whole manuscript.

7) In the Discussion, the authors claim that their work "provides insight into how mRNAs move between the piRNA pathway and the mutator complex and suggests a role for multiple perinuclear condensates to promote piRNA-mediated mRNA silencing". This has not been demonstrated. There are no experiments showing that the mRNAs targeted by piRNAs move through perinuclear SIMR-1 condensates to Mutator foci, nor that SIMR-1 is required for piRNA silencing. The reduction in a subset of piRNA targets might be an indirect consequence of SIMR-1 function, which still remains unknown. Thus, I suggest to remove this sentence. Same for sentences in the last paragraph of the subsection “RSD-2 and SIMR-1 promote the interaction between distinct primary and secondary siRNA pathways”.

8) In the Materials and methods section the authors should add the detailed protocol used for Immunoprecipitation and mass spec (buffer used etc.). The referenced Uebel et al. do not contain such information. Also, the raw proteomic data has not been released. Moreover, they should add a full description of the small RNA analysis performed as mentioned above.

[Editors’ note: further revisions were suggested prior to acceptance, as described below.]

Thank you for resubmitting your work entitled "A Tudor domain protein, SIMR-1, promotes siRNA production at piRNA-targeted mRNAs in *C. elegans*" for further consideration by *eLife*. Your revised article has been evaluated by James Manley as the Senior Editor, and Oliver Hobert as the Reviewing Editor.

The reviewers appreciate the revisions. Before we can accept the manuscript, we ask you to please consider some of the editorial revisions that both reviewers have raised and that appear reasonable. No further experimentation is required. We do not expect the revised paper to go back to the reviewers.

*Reviewer #1:*The authors have addressed our concerns with new experiments that support the main thesis of the paper: SIMR-1 functions in the piRNA silencing pathway.

*Reviewer #2:*In this revised manuscript, Manage et al. have addressed most of the major points I have raised in my review. Specifically, they have included data showing that SIMR-1 protein with mutation in a critical residue for the structure of the extended Tudor domain fails to localize to perinuclear granule and affects the fertility of the animal at 25°C, similar to the loss of function allele. Moreover, they have placed the reduction in small RNAs upon mutation of SIMR-1 in the context of piRNA pathway. Indeed, the mutant protein fails to silence a piRNA reporter transgene, in accordance to what I previously asked to show. They have also changed the text according to my suggestion and added more statistical tests (where it was possible) upon my request. However, I do have some additional comments that the authors should address:

1) Even if the Authors have removed all statements that "SIMR-1 "hands-off" RNAs from the piRNA to mutator pathway" in the main text, the passage in the Abstract that says "However, how mRNAs first recognized by the piRNA pathway are then handed off to the mutator pathway is unclear. Here, we identify the Tudor domain protein, SIMR-1, as acting at this step between piRNA production and mutator complex-dependent siRNA biogenesis" is still confusing, suggesting that SIMR-1 acts at that step.

2) The Authors have shown that there is a gradual reduction in small RNAs in *simr-1* mutant compared to WT across generations at 25°C in the new Figure 5—figure supplement 1D. However, the heat map shown in Figure 5C suggests that the reduction in small RNAs (even at generation 10) in *simr-1* mutant is milder compared to *prg-1* mutant. Therefore, I recommend to include in Figure 5—figure supplement 1D data from *prg-1* mutant at 20°C and to calculate the significant changes in siRNAs between *prg-1* and *simr-1* mutants. If the small RNA depletion in *prg-1* mutant is more severe than in *simr-1* mutant (as it appears from Figure 5C), then the Authors should reconsider their claim that SIMR-1 act downstream the piRNA pathway to produce siRNAs in mutator foci. SIMR-1 might cooperate to accumulate small RNAs downstream of piRNAs together with other pathways or proteins (acting together or independent of SIMR-1 in piRNA-dependent small RNA accumulation). Also, the effect on small RNA accumulation can be still indirectly linked to piRNA-dependent small RNA biogenesis, since no interaction between SIMR-1 and PIWI or MUT-16 has been shown by Co-IP. Therefore, these alternative possibilities should be discussed in the text.

3) The increased siRNAs in simr-1 mutant also appear to be less severe than piRNA mutant (Figure 5C). Similarly, the changes in mRNAs (including histone mRNAs) might be less severe that piRNA mutant. In my opinion this can be an important determinant in explaining the different phenotype between piRNA and *simr-1* mutants.

4) In the paragraph now called "*simr-1* Mrt phenotype is associated with increased levels of germ cell apoptosis" the Authors speculate that the significant increase in apoptotic cells at later generation in *simr-1* mutant compared to the relative WT is correlated to sterility. As I have already mentioned in my first review, although WT animals at 25°C have levels of apoptotic cells similar to those of later generation of sterile simr-1 mutant at 25°C, yet they are fertile. Thus, the number of apoptotic cells detected with this assay in simr-1 mutant do not suggest that they contribute to sterility. As a comparison, mutants of H3K9me (met-2, set-25) show 4-fold increases in apoptotic cells at 25°C compared to WT (in Zeller et al., 2016, Figure 2B). Therefore, in my opinion this data is inconclusive and can induce to misinterpretation. This is why, if the Authors are willing to show this result, they should comment on the fact that WT animals with similar number of apoptotic cells are fertile and that the number of apoptotic cells detected in *simr-1* mutant might not contribute to the sterility.

5) In the Introduction, the Authors wrote: "Here we identify a protein required to mediate the interaction between PRG-1 and piRNAs in P granules with siRNA amplification in Mutator foci." The interaction between these factors has not been confirmed by Co-IP or other experiments, thus the statement needs to be changed.

6) In the Introduction, the Authors claim that "this work identifies one of the first factors to act downstream of PRG-1 to mediate the production of secondary siRNAs". This is not accurate because many factors involved in the accumulation of piRNA-dependent small RNAs have been identified thus far. Also, the result presented pointed out that *simr-1* might directly or indirectly participate in the accumulation of those siRNAs (see also comment 2).

---

## [Author Response]

[Editors’ note: the authors resubmitted a revised version of the paper for consideration. What follows is the authors’ response to the first round of review.]

Reviewer #1:[…] The manuscript by Manage et al., reports on a new component of the *C. elegans* small RNA synthesis machinery. SIMR-1 contains a TUDOR-like domain, a domain often found in Argonaute-interacting proteins. Proteomic analyses of SIMR-1 indicate that SIMR-1 interacts with MUT-16, other Mutator foci proteins, and WAGO-1, an Argonaute that also interacts with MUT-16. Like Mutator foci proteins, SIMR-1 is required for the synthesis and/or maintenance of small RNAs targeting many loci (>600), including ~100 loci also targeted by piRNAs. SIMR-1 localizes to a new granule adjacent to Mutator foci. The authors speculate that SIMR-1 may function in the "handoff" of mRNAs targeted by piRNAs to mutator foci. This conclusion, however, does not seem well supported by the data since SIMR-1 mutants affect many more genes beyond those targeted by piRNAs (point #1). SIMR-1 interacts with the Argonaute WAGO-1, but a functional connection between SIMR-1 and WAGO-1 was not explored (point #5).

We have removed all statements that SIMR-1 “hands-off” RNAs from the piRNA to *mutator* pathway and have instead indicated that SIMR-1 acts at a step between piRNA production and *mutator*-complex-dependent siRNA biogenesis. To provide additional evidence that SIMR-1 acts in the piRNA pathway and not the downstream *mutator* pathway, we have included data showing that *simr-1* mutants, like mutants in the piRNA pathway, de-silence a sensitized piRNA sensor but not a piRNA sensor silenced by RNAe or an ERGO-1 dependent siRNA sensor (Figure 4A and Figure 4—figure supplement 1A-B). Furthermore, SIMR-1 is required to promote fertility after re-establishing production of WAGO-class 22G-siRNAs, a phenotype also associated only with mutants in the piRNA pathway (Figure 4B).

1) The authors do a comprehensive analysis of small RNA defects in SIMR-1 mutants raised at different temperatures and identify nearly ~1000 affected loci. Consistent with SIMR-1 interacting with the Mutator scaffold MUT-16, >90% of the genes affected in simr-1 mutants are also affected in mut-16 mutants. These genes include genes targeted by piRNAs, but the latter represent only a minority (<20%) of genes affected in simr-1 mutants, so it is not clear why the authors focus so much on those genes.

In the first version of this manuscript, we used Supplementary Table 2 from Lee et al., 2012, to define piRNA targets. This list contains approximately 300 genes for which the authors observed 8-fold enrichment of 22G-siRNAs in FLAG::PRG-1 compared to *prg-1* mutants and is likely not a comprehensive list of *prg-1*-dependent 22G-siRNA target loci. Therefore, we generated new small RNA libraries from wild-type and *prg-1* mutants at 20°C and defined *prg-1*-dependent siRNA target genes as those with at least 2-fold reduction in total small RNAs and at least 10RPM in wild-type or mutant libraries (the same criteria by which we defined *simr-*1-dependent siRNA target genes). Using these new libraries, we generated a more *comprehensive* list of 1758 *prg-1*-dependent siRNA target genes. With this larger list, now 686 (out of 817) or 84% of *simr-1*-dependent siRNA target genes overlap with *prg-1*-dependent siRNA target genes at 20°C (Figure 5D). At 25°C, 693 (out of 1258) or 55% of *simr-1*-dependent siRNA target genes overlap with *prg-*1-dependent siRNA target genes (Figure 5—figure supplement 1C). It is important to note that in the latter case the *simr-1* libraries, collected at 25°C, are being compared to *prg-1* libraries collected at 20°C and this difference in temperature may account for the lower overlap between *simr-1* and *prg-1*-dependent siRNA target genes. At both temperatures, the majority of *simr-1* targets (94% at 20°C and 80% at 25°C) still overlap with *mut-16*-dependent siRNA target genes (Figure 5D and Figure 5—figure supplement 1C).

We have also, since the first submission, obtained two alleles of *simr-1* out of a screen to de-silence a piRNA sensor strain (Figure 4A) and demonstrated that SIMR-1 is required to promote fertility after re-establishing WAGO-class 22G-siRNA production, a phenotype previously observed only with mutants in known piRNA pathway factors *prg-1*, *pid-1*, and *henn-1* (Figure 4B), providing further evidence that SIMR-1 is important for production of siRNAs at piRNA-target loci.

2) 'simr-1 mutants fail to produce high levels of siRNAs from the majority of piRNA-target loci' Figure 4C seems to contradict this statement since at 20°C only <50% of piRNA-target loci are affected. Also, what criteria were used to define "piRNA-targeted loci"? New analyses by the Mello and Lee labs have identified piRNA binding sites in 100s of mRNAs targeted by PRG-1/piRNA complexes. What happens to those loci in simr-1 mutants? Are the small RNAs completely gone or reduced? Are small RNAs nearest piRNA-binding sites most or least affected? is there a specific pattern of loss?

This sentence was changed to say that *simr-1* mutants fail to produce high levels of siRNAs at many piRNA-target loci. See previous response for more detail on how we defined piRNA-target loci (those with at least 2-fold reduction in total small RNAs in *prg-1* mutants and at least 10RPM in wild-type or *prg-1* libraries). While there is a significant overlap between the genes we defined as *simr-1*-dependent siRNA loci and *prg-1*-dependent siRNA loci there are also many *prg-1*-dependent siRNA loci that do not meet the criteria that we used to define *simr-1*-dependent siRNA loci. Note, however, that many of these had reduced siRNAs in the *simr-1* mutant but did not meet the 2-fold cutoff we used (see boxplot in Figure 5B where more than 75% of piRNA target genes have reduced siRNAs in a *simr-1* mutant).

Regarding piRNA binding sites, we decided to focus specifically on piRNAs whose expression changes in a *simr-1* mutant (Figure 5F and Supplementary file 4). We identified 77 piRNAs (less than 1% of all piRNAs) that are at least 2-fold down-regulated in a *simr-1* mutant. Using piRTarBase (which compiles piRNA binding sites identified by the Mello and Lee labs) we identified 37 mRNAs that are predicted targets of these piRNAs by the piRScan relaxed targeting rules (Lee lab) and the CLASH data (Mello lab). We chose to examine sites that were in common between both of these methods because Wu et al., 2019, demonstrate that only when they examined the piRNA target sites in common between these two methods was it predictive of an increase in mRNA expression and a reduction in 22G levels in a *prg-1* mutant; genes identified by either method alone were not significantly predictive. Of these 37 mRNAs that are predicted targets of piRNAs affected in a *simr-1* mutant, only five have reduced levels of siRNAs in a *simr-1* mutant (see Supplementary file 4). These data indicate that the *simr-1*-depleted piRNAs are not a major driver of siRNA depletion in *simr-1* mutants.

3) The authors argue that the defect must be "downstream of the piRNAs" because piRNAs levels overall are not affected in SIMR-1 mutants. They don't report however the levels of the specific piRNAs that target the ~100 piRNA-targeted genes affected in the simr-1 mutant. The levels of PRG-1 protein (piRNA associated Argonaute) also are not reported.

See previous response. We first identified all piRNAs whose expression is reduced by 2-fold in *simr-1* mutants (77 total piRNAs) and then demonstrate that the predicted target mRNAs for these piRNAs are, overall, not depleted of small RNAs in the *simr-1* mutant. In total, only five of the 817 genes reduced of small RNAs by at least 2-fold are predicted to be direct targets of these 77 “*simr-1*-dependent” piRNAs. We have additionally added a western blot for PRG-1 showing that the levels of PRG-1 are not affected by the *simr-1* mutant (Figure 7—figure supplement 1C).

4) Do the small RNA changes observed in simr-1 mutants correlate with changes in mRNA levels?

The majority of the genes that are depleted of small RNAs in *simr-1* are not affected at the mRNA level. In contrast, the genes that are enriched for small RNAs tend to have reduced expression at the mRNA level (Figure 6H). The failure of the genes depleted of small RNAs to increase in expression at the mRNA level is consistent with the recent observations by Reed et al., 2019, looking at all *mutator* targets and Barucci et al., 2020, looking at piRNA targets. In both cases, the majority of genes that were depleted of small RNAs failed to increase in mRNA expression. This data indicates that there may be additional layers of gene regulation that maintain silencing of these genes in the absence of siRNAs.

5) SIMR-1 interacts with the Argonaute WAGO-1, but the authors do not explore potential functional connections between SIMR-1 and WAGO-1 – why? Is there a good overlap between WAGO-1-dependent loci and SIMR-1-dependent loci?

MUT-16 interacts via immunoprecipitation and yeast two-hybrid with WAGO-1, but we have no evidence that SIMR-1 interacts with WAGO-1. Therefore, we did not follow up on this possibility.

6) Figure 6: Systematic quantification of relative granule positions shown is missing. See Wan et al., 2018, for examples of quantification metrics. The authors' findings suggest that they are 4 different types of perinuclear condensates in germ cells – are they all present in a 1:1:1:1 proportion? Is every P granule next to a Z granule next to a SIMR-1 granule next to a mutator focus or does the order and composition vary?

We have added quantification showing the percentage of SIMR-1 foci that are associated with P granules, Z granules, and *Mutator* foci. Additionally, we have included quantification of the distances between fluorescence centers for each of these foci relative to SIMR-1 foci (Figure 7B) and pointed out that in some cases we can observe multiple SIMR-1 foci interacting with another type of granule (see Figure 7A, arrow in inset for SIMR-1 and ZNFX-1 localization). These data show that while SIMR-1 is usually associated with the other three granules, it is not necessarily in a 1:1:1:1 proportion. We have also noticed that not all P granules are associated with SIMR/Mutator/Z granules. We do not know at this time why only some P granules are associated with these other granules however we are quantifying and exploring further in another manuscript.

Reviewer #2:In this manuscript, Manage et al. have identified a previously uncharacterized protein, called SIMR-1, which interacts with a component of mutator foci, MUT-16, and participates in the accumulation of a subset of endo-siRNAs antisense to piRNA and mutator target genes. They also show that simr-1 mutant displays a temperature-dependent progressive loss of fertility similarly to mutation in downstream components of the piRNA pathway. Interestingly, they have shown that SIMR-1 has a perinuclear localization in a region distinct from p-granule, z-granule or mutator foci, and that its localization is independent from mutator foci. Moreover, they have identified by sequence alignment an extended Tudor domain protein which might potentially interact with methylated PIWI protein. Based on these results, they propose that SIMR-1 constitutes a distinct germ-granule that acts downstream of PIWI and piRNAs to transport mRNAs targeted by piRNAs from p-granule to mutator foci (the granule involved in the amplification of endo-siRNAs). Overall, the distinct perinuclear localization of this protein is exciting, however there are no experiments that substantiate the claim that this protein has a role in transporting mRNAs form piRNA pathway to downstream mutator foci nor that the putative Tudor domain is functional. I also express some concerns about the methods, data analysis, and interpretation of some key results as listed below.

We have removed all statements that SIMR-1 transports RNAs from the piRNA to *mutator* pathway and have instead indicated that SIMR-1 acts at a step between piRNA production and *mutator*-complex-dependent siRNA biogenesis. We have also added data demonstrating that the Tudor domain is functional (see response to point “B” below). Finally, we have addressed all specific comments about data analysis and would be happy to provide any other information, if requested by the reviewer.

A) The authors performed a MUT-16 IPs coupled with mass spectrometry to identify novel interacting factors of mutator complex. My first concern is related to the fact that the authors did not specify how many IP replicates they have used and which statistical methods they have performed to identify specific interactors in MUT-16 IPs versus control IPs. It appears that they have considered interacting proteins that have peptides only in MUT-16 GFP and FLAG IPs and not in the control. This approach can limit the number of interacting proteins identifiable in the IPs. Also, the number of peptides they have identified for SIMR-1 is extremely low (2 peptides in FLAG IPs and 1 peptide in GFP IPs) rising doubt that SIMR-1 is a real interactor of MUT-16. Because of the lack of statistical tools used to identify interacting proteins I suggest them to perform Co-IP experiments to confirm this interaction. It is also not known whether this interaction is direct or mediated by RNA, and an experiment such as Co-IP can help to distinguish the nature of their interactions. Also, in Figure 1A they have included the proteins RSD-2, WAGO-1, and MATH-33, which have been identified following another (not clear) approach (their peptides are not uniquely present in MUT-16 IPs). Therefore, it should be removed from this table. Again, if they want to take in consideration such cases they should perform a quantitative proteomic approach, using more replicates and statistical tools.

Only a single replicate was performed for each immunoprecipitation (MUT-16-FLAG, MUT-16-GFP, control-FLAG, control GFP) so we did not perform statistical analysis. The approach was treated as a screen, to identify candidate genes for further exploration, rather than a comprehensive list of true MUT-16 interactors. We have removed RSD-2, WAGO-1, and MATH-33 from the table in Figure 1, as requested. Because we did perform further analysis with these proteins (localization and small RNA seq) that appear in other figures, they were instead included in Figure 1—figure supplement 1, along with the peptide counts in the control strains to make clear that these three proteins were identified at much lower coverage in the control FLAG IP.

We have attempted to confirm the SIMR-1-MUT-16 co-IPs to demonstrate that this is a true interaction, however our results were inconclusive. We believe that this result suggests that the interaction between MUT-16 and SIMR-1 may be weak or transient, which is not surprising given their localization to distinct compartments and the fact that SIMR-1 is not required for exogenous RNAi or small RNA production at many mutator target genes.

B) There are no functional data or structural information to asses that the conserved residues contained in SIMR-1 constitute a Tudor domain capable of recognizing a methylated substrate such as PIWI. Moreover, SIMR-1 IP and mass spec experiments failed to identify PRG-1/PIWI protein. The major interactor identified by mass spec is instead RSD-2, which they show to colocalize in the same perinuclear foci as SIMR-1, although it appears to have a different function (RSD-2 is involved in RNAi and not 22G-RNA amplification from piRNA targets and SIMR-1 is not involved in RNAi but in the accumulation of 22G-RNAs from piRNA targets). Therefore, the distinct localization of SIMR-1 per se does not implicate that SIMR-1 mediates the handoff of mRNAs from the piRNA pathway to the downstream Mutator components. Also, the peptide counts of the putative interactors of SIMR-1 shown in Figure 6—figure supplement 1 are extremely low, a factor that rises concerns on the quality of the IP/mass spec experiments. Again, it is not specified how many replicates have been performed. Therefore, some of these interactions, such as MUT-16/SIMR-1 should be confirmed by Co-IP.

To demonstrate that the Tudor domain of SIMR-1 is functional, we made a single point mutation in a critical residue for the structure of the extended Tudor domain and found that SIMR-1 no longer localizes to perinuclear granules (Figure 4C). Furthermore, these mutant animals have transgenerational fertility defects at elevated temperature similar to the *simr-1* deletion allele (Figure 4D). This same allele was also identified in an unbiased screen for de-silencing of a piRNA sensor (Figure 4A). While this data does not prove that the SIMR-1 Tudor domain can recognize methylated peptides, it does support the identification of the Tudor domain and indicate its critical role in SIMR-1 function.

We have removed the SIMR-1 mass-spec experiment due to the low peptide counts and lack of replicates. We attempted to replicate the MUT-16-SIMR-1 interaction by co-IP, but as described in the previous point, the results were inconclusive. Nonetheless, SIMR-1 clearly has a role in the production of siRNAs from piRNA targets based on its phenotypes. We plan to revisit the direct interactors of SIMR-1 in future work.

C) The authors performed a small RNA-seq analysis to check whether specific population of small RNAs are affected in simr-1 mutant. They have performed profiles of small RNAs in different conditions (normal temperature of grow or 25°C at several generations), perhaps to correlate gene expression changes with the temperature-dependent progressive loss of fertility phenotype. However, the lack of corresponding RNA-seq profiles does not allow to investigate the effect of temperature on gene expression and how it correlates with the progressive sterility phenotype. Moreover, the most dramatic changes in small RNAs in SIMR-1 mutant occur at 25°C after just one generation. Thus, there is no correlation between progressive loss of fertility and changes in small RNAs. I wonder then if the results across generation of SIMR-1 mutant grown at 25°C are relevant for the current manuscript at all. Therefore, the only conclusion from all the analyses performed is that SIMR-1 is somehow participating in the accumulation of a subset of mut-16 and piRNA targets. The effect of the temperature on small RNA biogenesis is instead not clear.

We have added a panel showing a reduction in small RNAs mapping to piRNA target genes in *simr-1* mutants over the course of 10 generations at 25°C (Figure 5—figure supplement 1D). Additionally, we have added mRNA-seq data to the manuscript, in part, to address whether we can correlate any gene expression changes to the progressive sterility phenotype. We find that some *mutator* targets, PRG-1 targets, and spermatogenesis genes are up-regulated exclusively at late (sterile or nearly sterile) generations, however, similar classes of genes are also up-regulated at 20°C (Figure 6G). Thus, it is difficult to attribute the sterility phenotype to reduced or increased expression of a specific gene class. We do observe defects in histone mRNA expression (Figure 6I and Figure 6—figure supplement 1G) which Barucci et al., 2020) propose leads to sterility in *prg-1* animals; however, in *simr-1* mutants the reduction in histone mRNA expression is observed even at 20°C, which suggests that it cannot be the sole cause of the fertility defect. We therefore agree with the reviewer that much of the changes in small RNA expression occur even at 20°C or after a single generation at 25°C – we hypothesize that this loss of piRNA and *mutator*-target small RNAs is sensitizing the animals to fertility defects at elevated temperature but this may not be a direct effect.

D) In addition, the small RNA analysis they have performed is not very compelling. For instance, in some cases is not clear if the criteria they have used to define genes with decrease or increase small RNAs. Also, the methods don't clearly show which small RNAs they are considering (22G-RNAs, 26G-RNAs, sense, antisense, broad range of small RNAs 21-26nt, etc.). Because these are small RNAs cloned from total RNAs and not from AGO IPs, the authors cannot be sure that the increases or decreases of small RNAs observed belong to a specific AGO pathway (it is known that mutation in some small RNA biogenesis factor causes misrouting of small RNAs into AGOs). In general, it appears that mutation in SIMR-1 causes a general downregulation (not completely loss) of small RNAs, which are considered to be mut-16 dependent (including some previously defined piRNA targets). Among these affected genes, the piRNA targets appear more affected than the whole mut-16 targets.

We have clarified the figures and figure legends to indicate what classes of small RNAs we are examining. In general, we are looking at total small RNAs mapping to a particular gene class and have not restricted it by size, 5’ nucleotide, or strandedness. For piRNAs, we looked only at small RNAs mapping to annotated piRNA loci. We do not argue that these small RNAs are binding particular Argonaute proteins, only that they map to known *mutator*, piRNA, CSR-1, etc target genes. Finally, using our new *prg-1* mutant small RNA data, we show that the majority of *simr-1* targets are also piRNA targets, in addition to being *mutator* targets (Figure 5D).

E) The authors claim that SIMR-1 acts downstream of the piRNA pathway, however they haven't performed any experiment to prove that piRNA silencing is impaired in simr-1 mutant. They should use the canonical piRNA sensor strain (Ashe et al., 2012), and check whether the silencing of the piRNA sensor reporter is impaired in simr-1 mutant. This experiment goes well with the other assays they have used to show that RNAi is not affected in simr-1 mutant.

We have now added the isolation of two alleles of *simr-1* from a piRNA sensor screen (first described in de Albuquerque, Genes Dev 2014) to demonstrate that *simr-1* can de-silence a sensitized piRNA sensor (Figure 4A) but not a piRNA sensor silenced by RNAe (Figure 4—figure supplement 1A). Furthermore, we have demonstrated that SIMR-1 is required to promote fertility after re-establishing WAGO-class 22G-siRNA production, a phenotype previously observed only with mutants in known piRNA pathway factors *prg-1*, *pid-1*, and *henn-1* (Figure 4B), providing further evidence that SIMR-1 is important for production of siRNAs at piRNA-target loci.

Specific comments:1) Add in Figure 2C the statistical significance of those changes.

The libraries in Figure 2C were considered part of the screen to identify new small RNA factors and were not performed with replicates, so statistics was not included. All subsequent small RNA and mRNA seq focused on *simr-1* (Figures 5-6 and Figure 5—figure supplement 1 and Figure 6—figure supplement 1) were performed with multiple replicate libraries and include statistics.

2) Add in Figure 2D genes that are statistically significant up-regulated and down-regulated (or >2-fold change). They can label the significant dots with different colours as well. Same in 2E, if any.

Colors have been changed to highlight genes with log_2_(fold change) in small RNA abundance greater or less than 1.

3) In Figure 4C the total number of genes belonging to piRNA, ERGO-1, and mut-16 is changing in the different Venn diagram. This is not clear. Shouldn't be constant?

These particular Venn diagrams have been replaced, however in the original figure, overlaps of less than 10 genes were not shown on the diagram for simplicity.

4) Subsection “simr-1 mutants have increased levels of germ cell apoptosis”, the paragraph title is misleading. The level of germline apoptosis in SIMR-1 is similar to WT at 20°C and lower than WT at 25°C (first generation). It is slightly higher than the corresponding WT at generation 2, 10, 11 and not changing at generation 4,7. Importantly, the WT at 25°C (1 generation) has the highest level of germline apoptosis compared to all the other data points and yet is still fertile. Therefore, the reported level of germline apoptosis is not correlated with the sterility phenotype. This is why I disagree with the Authors' interpretation of the results shown in Figure 3E and the conclusion they wrote in the subsection “simr-1 mutants have increased levels of germ cell apoptosis”.

We have changed the title of this section to indicate that the increase in apoptosis is specifically associated with the Mrt phenotype in generations 10 and 11. We agree that it is strange that apoptosis increases so significantly in the first generation at 25°C, however we interpret this to be a response to the heat stress. By generation 2 at 25°C, the animals have acclimated to the elevated temperature and apoptosis is reduced. Only in the late generation 25°C animals do we observe apoptosis elevated in the *simr-1* mutant compared to wild-type. While we are not arguing that the apoptosis is causative of the sterility, we do believe that the increase in apoptosis in the late generations, which is far more significant than any differences observed in earlier generations, is correlated with reduced fertility in these animals.

5) The first paragraph of the subsection “SIMR-1 is required for small RNA production at piRNA target genes”, contains results that are not shown in any figures. Please, provide a figure that report these results. Also, it is not written which criteria they have used to identify changes in small RNA populations (fold changes, p-value?), nor it is known from the Materials and methods or text which small RNAs they are considering (22G-RNAs, 26G-RNAs, sense, antisense, broad range of small RNAs 21-26nt, etc.). Also, do the small RNA-depleted genes at 25°C (which are not depleted in subsequent generations) reacquire the small RNAs in progressive generations?

We have added tables to indicate the total number of genes with changes in small RNAs or mRNAs in each genotype and condition (Figure 5A and Figure 6F). Details regarding criteria used to define genes with small RNA and mRNA changes and types of small RNAs being analyzed has been added to appropriate figure legends and Materials and methods.

For the genes that lose small RNAs in *simr-1* mutants, the majority are found to lose small RNAs across multiple conditions (temperature or generation). For the genes that lose small RNAs in *simr-1* mutants only in the first generation at 25°C, but not at 20°C or later generations at 25°C (~300 genes in total), some of these genes have increased small RNAs in the wild-type strain only in this condition and are lost in subsequent generations while others lose small RNAs that are then regained at later generations. In the cases of genes that appear to gain small RNAs in *simr-1* mutants only in the first generation at 25°C, but not at 20°C or later generations at 25°C, many of these genes actually have reduced small RNAs in the wild-type strain at this generation that are recovered at later generations. In fact, there are many genes that gain and lose small RNAs in the wild-type strain at elevated temperature and across generations, but this analysis seemed beyond the scope of the current paper so we did not include the data.

6) The results from the analyses shown in the subsection “simr-1 mutants have increased levels of siRNAs mapping to histone genes”, are also not clear and maybe not relevant for the current manuscript. The results described lack a figure reference. Are they shown in Supplementary Figure 5B? Also, in Supplementary Figure 5B they only show depleted siRNAs and not enriched siRNAs. Also, they only consider lists of genes depleted of 22G-RNAs. Maybe if they consider log2(fold changes) they will find more insightful results. In regard to the small RNAs targeting histone mRNAs they have cited a submitted paper (not sure if it is possible to cite the journal where the paper is submitted) and conclude this might be relevant. Mutation in MUT-16, which is linked to piRNA pathway, do not show the same histone small RNAs. Moreover, histone small RNAs are present at all temperature, therefore also do not correlate with the progressive loss of fertility in SIMR-1 mutant. Again, I am not sure how this paragraph fits with the whole manuscript.

We believe that the histone data supports our argument that *simr-1* acts in the piRNA pathway (and not the *mutator* pathway). The paper referenced in the original submission of this manuscript is now published in Nucleic Acids Research (Reed et al., 2019). An additional paper has also recently been published on this topic (Barucci et al., 2020), both demonstrating that increased small RNAs mapping to histone genes and reduced histone mRNA expression is a phenotype associated with the piRNA pathway. Both papers also demonstrate that these small RNAs mapping to histones are dependent on the *mutator* complex for their biogenesis, thus if they are present, the *mutator* complex must be functional. Therefore, these data places *simr-1* in the piRNA pathway, upstream of the *mutator* complex-dependent biogenesis of these histone small RNAs.

The sentences which were lacking a figure reference have been removed. Supplementary Figure 5B has been removed and replaced with Figure 6B-C, which shows genes enriched for small RNAs in *prg-1*, *mut-16*, and *simr-1* mutants, and the correlation of those enriched for small RNAs in *prg-1* and *simr-1* mutants with histone genes. All of the lists of genes enriched for small RNAs are described in Figure 5A and Supplementary file 3.

7) In the Discussion, the authors claim that their work "provides insight into how mRNAs move between the piRNA pathway and the mutator complex and suggests a role for multiple perinuclear condensates to promote piRNA-mediated mRNA silencing". This has not been demonstrated. There are no experiments showing that the mRNAs targeted by piRNAs move through perinuclear SIMR-1 condensates to Mutator foci, nor that SIMR-1 is required for piRNA silencing. The reduction in a subset of piRNA targets might be an indirect consequence of SIMR-1 function, which still remains unknown. Thus, I suggest to remove this sentence. Same for sentences in the last paragraph of the subsection “RSD-2 and SIMR-1 promote the interaction between distinct primary and secondary siRNA pathways”.

Both sentences have been edited to remove language indicating trafficking or handover between the piRNA pathway and the mutator complex, and simply state that SIMR-1 acts downstream of piRNA biogenesis and upstream of mutator complex-dependent siRNA amplification.

8) In the Materials and methods section the authors should add the detailed protocol used for Immunoprecipitation and mass spec (buffer used etc.). The referenced Uebel et al. do not contain such information. Also, the raw proteomic data has not been released. Moreover, they should add a full description of the small RNA analysis performed as mentioned above.

The IP/mass spec section of the Materials and methods has been updated to include buffers used and the Bioinformatics Analysis section has been modified to include more detail about the packages used and the analysis performed. A table of all mass spec data has been included as Supplementary file 2. We would be happy to provide any additional details, if requested by the reviewer.

[Editors’ note: what follows is the authors’ response to the second round of review.]

Reviewer #2:[…] 1) Even if the Authors have removed all statements that "SIMR-1 "hands-off" RNAs from the piRNA to mutator pathway" in the main text, the passage in the Abstract that says "However, how mRNAs first recognized by the piRNA pathway are then handed off to the mutator pathway is unclear. Here, we identify the Tudor domain protein, SIMR-1, as acting at this step between piRNA production and mutator complex-dependent siRNA biogenesis" is still confusing, suggesting that SIMR-1 acts at that step.

We have edited the Abstract to eliminate the suggestion that a “hand-off” is occurring.

2) The Authors have shown that there is a gradual reduction in small RNAs in simr-1 mutant compared to WT across generations at 25°C in the new Figure 5—figure supplement 1D. However, the heat map shown in figure 5C suggests that the reduction in small RNAs (even at generation 10) in simr-1 mutant is milder compared to prg-1 mutant. Therefore, I recommend to include in Figure 5—figure supplement 1D data from prg-1 mutant at 20°C and to calculate the significant changes in siRNAs between prg-1 and simr-1 mutants. If the small RNA depletion in prg-1 mutant is more severe than in simr-1 mutant (as it appears from Figure 5C), then the Authors should reconsider their claim that SIMR-1 act downstream the piRNA pathway to produce siRNAs in mutator foci. SIMR-1 might cooperate to accumulate small RNAs downstream of piRNAs together with other pathways or proteins (acting together or independent of SIMR-1 in piRNA-dependent small RNA accumulation). Also, the effect on small RNA accumulation can be still indirectly linked to piRNA-dependent small RNA biogenesis, since no interaction between SIMR-1 and PIWI or MUT-16 has been shown by Co-IP. Therefore, these alternative possibilities should be discussed in the text.

We added a line in Figure 5—figure supplement 1D below which all piRNA targets fall (since the list of piRNA-target genes was defined as those reduced by 2-fold in a *prg-1* mutant). We agree that the reviewer makes a good point that piRNA-target genes are reduced of siRNAs in *prg-1* more severely than *simr-1*, even at late generations. Thus, we have adjusted the text to reflect that sterility is unlikely to be caused by this loss progressive loss of siRNAs in *simr-1*, as *prg-1* was fertile at the time it had an even greater loss of siRNAs to these genes. We have additionally modified the first paragraph of the Discussion to point out that the *simr-1* phenotypes are weaker than those of *prg-1*, suggesting that *simr-1* is not absolutely required to mediate siRNA amplification at all piRNA target genes, or it acts cooperatively with other pathways or proteins.

3) The increased siRNAs in simr-1 mutant also appear to be less severe than piRNA mutant (Figure 5C). Similarly, the changes in mRNAs (including histone mRNAs) might be less severe that piRNA mutant. In my opinion this can be an important determinant in explaining the different phenotype between piRNA and simr-1 mutants.

As mentioned in previous comment, we have edited the first paragraph of the Discussion to discuss the difference in phenotypes between *simr-1* and *prg-1* and offer possible reasons for this.

4) In the paragraph now called "simr-1 Mrt phenotype is associated with increased levels of germ cell apoptosis" the Authors speculate that the significant increase in apoptotic cells at later generation in simr-1 mutant compared to the relative WT is correlated to sterility. As I have already mentioned in my first review, although WT animals at 25°C have levels of apoptotic cells similar to those of later generation of sterile simr-1 mutant at 25°C, yet they are fertile. Thus, the number of apoptotic cells detected with this assay in simr-1 mutant do not suggest that they contribute to sterility. As a comparison, mutants of H3K9me (met-2, set-25) show 4-fold increases in apoptotic cells at 25°C compared to WT (in Zeller et al., 2016, Figure 2B). Therefore, in my opinion this data is inconclusive and can induce to misinterpretation. This is why, if the Authors are willing to show this result, they should comment on the fact that WT animals with similar number of apoptotic cells are fertile and that the number of apoptotic cells detected in simr-1 mutant might not contribute to the sterility.

An additional sentence has been added to this section of the Results to make it clear that high levels of apoptosis are not always correlated to sterility.

5) In the Introduction, the Authors wrote: "Here we identify a protein required to mediate the interaction between PRG-1 and piRNAs in P granules with siRNA amplification in Mutator foci." The interaction between these factors has not been confirmed by Co-IP or other experiments, thus the statement needs to be changed.

We have edited this sentence so as not to imply a direct physical interaction between SIMR-1 and these other pathways.

6) In the Introduction, the Authors claim that "this work identifies one of the first factors to act downstream of PRG-1 to mediate the production of secondary siRNAs". This is not accurate because many factors involved in the accumulation of piRNA-dependent small RNAs have been identified thus far. Also, the result presented pointed out that simr-1 might directly or indirectly participate in the accumulation of those siRNAs (see also comment 2).

We have edited this sentence so as not to indicate that SIMR-1 is the first factor identified with this role.